# Earthquake preparedness among religious minority groups: The case of the Jewish ultra-Orthodox society in Israel

Zvika Orr[1], Tehila Erblich[1], Shifra Gottlieb[1], Osnat Barnea[2], Moshe Weinstein[3], Amotz Agnon[2]

[1]Department of Nursing, Jerusalem College of Technology, Jerusalem, 9116001, Israel

[2]Neev Center for Geoinfomatics, Fredy & Nadine Herrmann Institute of Earth Sciences, The Hebrew University of Jerusalem, Jerusalem, 9190401, Israel

[3]Department of Electro-Optics Engineering, Jerusalem College of Technology, Jerusalem, 9116001, Israel

*Correspondence to:* Amotz Agnon (amotz@mail.huji.ac.il)

**Abstract.** To work effectively, emergency management systems that deal with earthquake threats must consider the needs of religious minority groups. Studies regarding earthquake preparedness among marginalized social-cultural groups can highlight ways to improve it. Recently, some research has focused on the effect of religion on earthquake preparedness. However, very few studies have connected the two and examined earthquake preparedness among religious groups that are also a social-cultural minority in relation to the authorities. This study examines the effects of religious beliefs and customs on earthquake preparedness among the Jewish ultra-Orthodox community in Israel, a significant religious minority with unique social, cultural, and economic characteristics. Data were obtained using mixed methods including a survey, in-depth interviews, and focus groups. Results demonstrated that the majority of the community had a low level of hazard knowledge and a high level of disbelief that a devastating earthquake would occur in their area in the near future. This is despite a long-documented history of earthquakes that devastated the Levant and, in particular, dwelling locations for this community. Low exposure to media, insularity of educational institutions, and suspicious attitudes toward state authorities were shown to hinder preparedness, while strong social capital improved it. This research is unique for it studies a religious group that is also a cultural minority, which, therefore, requires special adaptations. Some of the recommended adaptations include receiving support from religious leaders, publishing preparation guidelines in proper settings, working with civilian organizations that are seen as legitimate by the religious communities, and adapting technologies and information to be religiously appropriate. To conclude, this research offers a perspective on the complex reality of hazard preparedness in a religiously diverse country. The conclusions are applicable to other countries and natural hazards.

**1 Introduction**

Earthquakes often occur with little to no warning and have the potential to cause enormous amounts of destruction and death. The damage is mainly due to the collapse of man-made structures, such as buildings. Therefore, the disaster is not only the result of the severity of the earthquake ground motion (for example, magnitude, depth, and distance), but also of the distribution and size of the population and the degree of earthquake preparedness (Bertero and Bozorgnia, 2004; Mesgar and Jalilvand, 2016; Takagi and Wada, 2018).

Many countries have developed emergency management systems to address this threat. These systems must consider the needs of minority groups to work effectively for the society as a whole. Some research regarding response to disasters among marginalized social-cultural groups has been offering ways to improve preparedness (e.g., Maldonado et al., 2016; Shapira et al., 2018; Uekusa, 2019; Zhang, 2020). However, only few studies have focused on religious minority groups (e.g., Gianisa and Le De, 2018; Ngin et al., 2020), despite the fact that religious characteristics, in some cases, clearly have an impact on emergency preparedness.

The goal of our research is to examine the effect of religious beliefs and minority status on earthquake preparedness and recommend ways for improving risk mitigation in religious minority communities. To achieve this goal, we studied the Jewish ultra-Orthodox community in Israel, a significant religious minority in Israeli society that stands out with unique social, cultural, and economic characteristics. The research questions include the following: What is the level of preparedness in the Jewish ultra-Orthodox community? Which characteristics have a positive effect on preparedness and which have a negative effect? How can the level of preparedness and conduct during an emergency be improved?

The findings establish that religious minority groups have many characteristics and worldviews that significantly influence all stages of disaster response. Policymakers must take these features into consideration when attempting to upgrade the preparedness of religious groups in their country. This case study adds a unique perspective to the study of earthquake preparedness, which can help upgrade preparedness in other societies worldwide.

**2 Literature Review**

**2.1 Disaster Preparedness in Religious Communities and Minority Groups**

In recent years, studies have examined the influence of religion on disaster preparedness and response (e.g., Baytiyeh and Naja, 2014; Gianisa and Le De, 2018; Sun et al., 2018, 2019). These studies have shown that religion plays a crucial role in all stages of a disaster, affecting different aspects positively and negatively. Hence, scholars emphasize the importance of considering religious factors and beliefs when attempting to improve disaster management. These factors include the religious community's shared values, traditions, worldviews, goals, strengths, and vulnerabilities (Appleby-Arnold et al., 2018). Religion can impact a disaster from the preparation phase through the emergency phase and extend through the restoration phase.

Regarding **the preparation phase**, fatalistic beliefs, which are common in many religious societies, may lead to a passive attitude regarding the need for disaster preparedness (Baytiyeh and Naja, 2014; Sun et al., 2018,

2019). Plapp and Werner (2006) demonstrated that in some religious traditions, natural disasters are seen as a
divine punishment that cannot be prevented; consequently, none of the preparations called for by the civil
authorities are necessary (cf. Sun et al., 2019). A study on disaster preparedness in Martinique found that though
an earthquake awareness campaign succeeded, there was still a gap between the increased awareness and actual
level of preparedness. This gap can be explained at least in part by religious factors, such as the dominant
fatalism embedded in religious and magical beliefs (Audru et al., 2013; cf. Azim and Islam, 2016 in the context
of Saudi Arabia). Moreover, some religious communities have minimal exposure to media, which can increase
their vulnerability (Ya'ar et al., 2015). However, Gianisa and Le De (2018) showed how religious beliefs can
improve disaster preparedness. Muslims, among other denominations, believe that human destiny can be
changed by doing good deeds in this life, even though their fate has been predetermined before they were born.
This belief encourages people to prepare for a disaster and refrain from being fatalistic.
While attempting to improve disaster preparedness, it is crucial to contemplate religious factors and beliefs. For
example, in cultures where fatalistic and submissive attitudes prevail, critical thinking should be emphasized in
the educational frameworks (Baytiyeh and Naja, 2014; Yari et al., 2019). In the context of Muslim societies,
Azim and Islam (2016) proposed to include certain interpretations of the Quranic sources to support risk
mitigation strategies. With regard to the preparation guidelines, it is pertinent to choose proper agents to deliver
the information and adjust the content and transformation methods (Mileti et al., 2006). Some suggest
collaborating with the religiously affiliated community organizations, which often have a strong impact on all
areas of life in a religious society (Baytiyeh, 2017; Gianisa and Le De, 2018), and partnering with trusted
community leaders (Gil-Rivas and Kilmer, 2016). Similarly, Audru et al. (2013) illuminated the need to anchor
campaign efforts in local culture and religion, using the local language and knowledge, and by developing
educational techniques tailored to the needs of specific groups. Preventative training activities and information
campaigns in the education system are some of the most common and effective strategies in disaster risk
mitigation (Lucini, 2014; Smawfield, 2013). Last, decentralizing disaster risk reduction policies and measures,
thereby increasing the role of local government in decision-making, is also effective in improving preparedness
(Grady et al., 2016).
A growing body of evidence has established that appropriate behavior during **the emergency phase** can save
many lives and reduce property damage. In the Italian context, Lucini (2014) showed how cultural approaches
of diverse social groups can affect their behavior. Studies have shown that religion can have a major impact on
the emergency phase. For example, religious fatalistic beliefs may lead to passivity during the emergency
(Baytiyeh and Naja, 2014; Sun et al., 2018). On the positive side, religious beliefs and practices may serve as a
vital source of spiritual support in crucial times by bonding people together and helping them cope with the
disaster successfully (Gianisa and Le De, 2018; Sun et al., 2018). Gianisa and Le De (2018) described that
following the 7.6-magnitude earthquake that hit Padang city in Indonesia in 2009, prayer positively improved
people's coping abilities by giving them the reassurance that every event is caused by God, a belief that provided
them with the strength to face the event. Moreover, a cohesive religious community often has a strong social
network and social capital, which can optimize emergency management. Following an earthquake, for example,
community members will take action even before the emergency forces arrive (Aldrich, 2011; Gianisa and Le
De, 2018). During the emergency phase, it is vital for the state to support and coordinate emergency response at
the local level, since local response is crucial at this stage of a disaster (Alexander, 2010).
The **restoration phase** refers to resilience, which is the ability of communities and individuals to cope relatively
well during and after a crisis. Resilience includes the ability to rebuild, recover, and secure livelihoods, and
return quickly to normalcy. Aldrich (2011, 2012) maintained that the resilience of a community primarily
depends on its social capital or, in other words, the resources embedded in its social networks. He showed that
social capital acts as the strongest predictor of population recovery after a catastrophe, for example, by
providing informal insurance and housing (see also Wilkin et al., 2019). This factor is particularly significant in
many religious communities that have very strong social networks, such as the Jewish ultra-Orthodox
community in Israel (Caplan, 2007; Caplan and Stadler, 2012). As Sun et al. (2018) have explained, a shared
belief system can generate social capital and network-linked advantages, which can become a vital source of
support. This strength ought to be leveraged in the restoration, reconstruction, and rehabilitation phase.
Ngin et al. (2020) found that Cambodian and Thai Buddhist temples in Auckland, New Zealand have a pivotal
role in assisting their members to recover from natural hazard events. These researchers contended that faith,
culture, and social bonding are the main factors that allow the temples to perform their role. These findings
support the assertion that religious institutions' social capital plays a decisive role in urban resilience. Likewise,
in the context of Tibetan Buddhism, Sun et al. (2019) found that temples and the Buddhist clergy significantly
helped Tibetans cope with an earthquake disaster. These researchers showed that belief in Tibetan Buddhism
had a detrimental and constructive effect on the Tibetans' behavior. In the context of the 2005 earthquake in
Pakistan, Cheema et al. (2014) found that mosques have been entry points to access communities during
response and relief. One mosque even facilitated public and private sector activities during the recovery and
restoration phase. Mosques had a similar role in the aftermath of the 2004 Indian Ocean Earthquake and
Tsunami in Aceh, Indonesia (McGregor, 2010). These findings underscore the need to engage community-based
religious institutions and their leaders in the recovery phase and create partnerships between them and the
international organizations to win the local communities' trust and use local knowledge and resources (Cheema
et al., 2014). Scholars have also highlighted several factors that impede the potential contributions of
community-based religious institutions, such as gender inequality and political controversies (Cheema et al.,
2014), language barriers, generational divides, and internal divisions (Ngin et al., 2020). Furthermore, religious
beliefs can be an obstacle in relocating people, building back differently, or making other changes to livelihoods
to reduce exposure to future hazards (Schipper et al., 2014).
Religious groups may also be minorities in their countries, and religion and religiosity often intersect with
socioeconomic status, as clearly demonstrated by the case examined in this study. Therefore, it is pertinent to
examine the marginalization of certain religious groups, which can greatly impact earthquake preparedness and
raise vulnerability. Maldonado et al. (2016) found that minority groups, such as Hispanic immigrants in the
United States, show a low level of self-protection and preparedness, low level of hazard knowledge, and high
level of risk perception, all of which reflect a high degree of vulnerability. It is generally accepted that racial and
ethnic minorities and the poor are hit harder during crises and disasters are disproportionately debilitating for
disadvantaged social groups, which, therefore, need better preparedness (Lucini, 2014; Maldonado et al., 2016;
Shapira et al., 2018; Spence et al., 2007). Marginalized groups have a harder time accessing information and,
consequently, are at greater risk before and after a disaster (Kellman, 2011). Moreover, the guidelines
transferred by the mass media may ignore the needs of minority groups, which can lower the level of
compliance, such as in the case of COVID-19 in Israel (Kalagy et al., 2020; Waitzberg et al., 2020). In addition,
owing to their distrust in the government, minority groups are less likely to take the risk warning messages
seriously without confirming them first, especially if these messages were sent out directly by the government
(Spence et al., 2007). To conclude, religious groups are often vulnerable due to intersecting factors such as
poverty and minority status, like the Jewish ultra-Orthodox society in Israel.
**2.2 The Jewish Ultra-Orthodox Society**
In 2019, Israel's ultra-Orthodox population reached 1,125,000 million people, comprising 12.5% of the country's
total population (Cahaner and Malach, 2019). Ultra-Orthodox Jews adhere to a strict interpretation of Jewish
religious law. Religious precepts regulate all aspects of their daily life. Their life values, educational
frameworks, and culture distinguish them from all the other groups in Israeli society. The ultra-Orthodox Jews
strive to create a "scholarly society" where men are totally immersed in studying Jewish religious law. Some
other values and norms of the Jewish ultra-Orthodox society include a family-centered lifestyle, insular
communal life, conservatism, extensive social control of members' behavior, special dietary laws, gender
segregation, strict dress codes, and respect for the leadership of prominent rabbis in all areas of life. The Jewish
ultra-Orthodox society is characterized by high population growth, poverty, very limited participation in the
army draft, and a separate education system that focuses on religion, with a minimal presence of secular
subjects. Though many tend to view the ultra-Orthodox population as homogeneous, it is in fact composed of
several different communities belonging to different factions that diverge in their worldview, lifestyle, custom,
religious leadership, and economic and political institutions (Cahaner and Malach, 2019; Caplan, 2007; Caplan
and Stadler, 2012; Gal, 2015; Vardi et al., 2019). Most of the ultra-Orthodox people are Hebrew speakers,
although some of the communities still speak Yiddish as their first language (Assouline, 2017). Most ultra-
Orthodox people live either in towns of their own or in closed community neighborhoods within diverse cities
(Regev and Gordon, 2019; Shahak, 2017) (Figure 1). Recently, there has been a rise in ultra-Orthodox
participation in the general Israeli economy, society, and civic affairs. This change indicates that a large number
of ultra-Orthodox individuals believe that the society should move away from being a "scholarly society" where
men are discouraged from joining the workforce. However, this is not the view of the entire ultra-Orthodox
population (Caplan and Stadler, 2012; Vardi et al., 2019).
Regarding earthquake preparedness, many Jewish ultra-Orthodox people in Israel live in high-density
neighborhoods and buildings that do not meet the standards for earthquakes. According to the Central Bureau of
Statistics (2019), the density of the Jewish ultra-Orthodox population is 1.35 persons per room and that of the
Jewish secular population is 0.71 persons per room. In Jerusalem, where most of our interviews were conducted,
many of the neighborhoods with a majority of ultra-Orthodox residents (Figure 2) do not have earthquake-
resistant buildings since they were built before 1980 (Municipality of Jerusalem, 2020), the year in which a
seismic building code was introduced in Israel. As most earthquake fatalities result from building collapses (e.g.,
Coburn et al., 1992), the ultra-Orthodox population is subject to a considerable earthquake hazard.
As indicated above, besides having unique religious features, the Jewish ultra-Orthodox society is also a
minority group in their country. As the above literature review indicates, the responses of religious communities
and minority groups to natural disasters have been studied to some extent. Yet, to date, very few studies have
researched disaster preparedness among religious groups that also constitute a marginalized minority. It seems
that this intersection of religion and minority status may create distinct needs in terms of disaster risk reduction
— needs that thus far have not been sufficiently addressed. As a religious minority, the Jewish ultra-Orthodox
population is more vulnerable to disasters and demands specific adjustments in terms of disaster management.
Our research proposes to improve this population's preparedness and resilience by examining the community's
characteristics relevant to these matters. We believe that this case study can also be relevant to other religious
societies.

**2.3 Earthquake Hazard in Jerusalem**

Some of the ultra-Orthodox communities in Israel have lived in the vicinity of seismogenic faults for centuries,
and suffered severe damage and loss of life. Such is the case for Zefat (Figure 1) that was devastated in the 1759
and 1837 earthquakes (Katz and Crouvi, 2007). We focus our attention on Jerusalem where most of our
interviews were conducted. Throughout Israel, the active Dead Sea fault system (DSFS), which separates the
Arabian plate from the Sinai-Levant Block (Figure 1) (e.g., Quennell, 1958; Freund et al., 1968; Garfunkel,
1981), exposes the nearby settlements to earthquakes. Since 1900, a number of medium-to-strong earthquakes
have caused numerous casualties and extensive structural damage along the DSFS. On February 11, 2004, a
small, 5.1-magnitude earthquake occurred on the DSFS northeast of the Dead Sea (Figure 1). It caused minor
damage and panic in Jerusalem, 32 km to the west (Hofstetter et al., 2008). The 6.2-magnitude earthquake of
1927 centered north of the Dead Sea (e.g., Avni, 1999; Hough and Avni, 2011; Shapira et al., 1993), killed
several hundred people, injured a thousand or more, and destroyed numerous buildings (Figure 1). Since 1927,
the population has grown significantly, and so has the number of buildings. Clearly, such an earthquake would
lead to far greater damage today. In 1995, a strong 7.2-magnitude earthquake occurred on the DSFS, much
farther south in the Gulf of Aqaba (Figure 1). This earthquake was accompanied by more than 10 reportedly felt
aftershocks, causing damages to buildings in the towns of Aqaba (Al-Tarazi, 2000) and Eilat (Baer et al., 2008;
Shamir et al., 2003). This latest strong earthquake on the same DSFS is a potent reminder that a nearby
earthquake would have devastating effects. The population of Jerusalem and its buildings repeatedly have
suffered from such earthquakes on the DSFS (Ambraseys, 2009; Guidoboni & Comastri, 2005) (Figures 1, 2).
The ever-growing ultra-Orthodox neighborhoods are prone to increased ground shaking during earthquakes
(Figure 2).
Table 1 lists historic earthquakes that damaged Jerusalem since Roman times. The locations and intensities of
earlier catastrophes, depicted in the Bible or by Josephus Flavius are less certain (Ambraseys, 2009; Guidoboni
& Comastri, 2005). The sources of most of these events were likely in the Jordan Valley-Dead Sea sector of the
fault system (Figure 1) (Agnon, 2014; Lefevre et al., 2018).

**3 Methods**

In this research, we used a mixed methods approach that combines quantitative and qualitative methods. This approach allowed us to consider different aspects of the research field from multiple perspectives (Creswell and Plano Clark, 2011). Specifically, this study used convergent parallel mixed methods (Creswell, 2014). The quantitative aspect of the study included closed-ended questions in a survey, and the data were analyzed using statistical tools. The qualitative aspect included a few open-ended questions in a survey, in-depth interviews with key stakeholders, and focus groups with several organizations. Findings from the qualitative data were analyzed using qualitative content analysis and grounded theory, including an iterative search for repeating concepts and ideas (Corbin and Strauss, 2014).

## 3.1 Survey

We conducted a special-purpose social survey comprising both open-ended and closed-ended questions, which was distributed throughout the general Jewish ultra-Orthodox public. The questionnaire was based on a reliable and validated questionnaire created by Ya'ar et al. (2015), who studied the Israeli public's attitudes toward earthquake preparedness by asking the respondents to indicate the perceived level of earthquake risk and their self-assessed level of preparedness. We edited, processed, and adapted the questionnaire to the ultra-Orthodox society and distributed it to people through in-person interviews. Most of the interviews took place in Jerusalem and its neighboring towns, allowing us a deeper understanding of the residents' needs and challenges and enabling the possibility of working with them over time to upgrade preparedness. Subsequently, we created a shortened version of the questionnaire that was administered via an online form. The questionnaires received 228 responses: 140 through in-person interviews and 88 via the online form. The response rate was around 90% for the in-person interviews. The response rate for the online questionnaire is undefined since it was distributed via a free link (Appendix A).

The answers to the closed-ended questions were analyzed using statistical methods with SPSS (Statistical Package for the Social Sciences). The data collected using the open-ended questions was thematically analyzed both deductively and inductively according to key categories and themes.

The research questionnaire included the following topics: Sociodemographic features (age, civil status, area of living, socioeconomic status, community affiliation); involvement in earthquake preparedness activities; religion and preparedness; preparation in the respondent's home for an earthquake; subjective views on earthquake preparedness; coping capacities; and exposure to information regarding earthquake preparedness.

## 3.2 In-Depth Interviews and Focus Groups

The research team conducted 31 in-depth, semi-structured interviews based on an interview guide (qualitative questionnaire), without strictly adhering to it (Appendix B). The semi-structured interviews allowed us the flexibility and interactivity of qualitative research, while simultaneously facilitating a greater degree of standardization than more open "field" interviews (Kelly, 2010). The interviewees consisted of 16 relevant national-level policy and decision makers (e.g., from the Home Front Command, Ministry of Health, Ministry of Education, National Steering Committee for Earthquake Preparedness, etc.), 10 officials in rescue organizations, and five religious leaders and key figures in the ultra-Orthodox community. Of the interviewees, 17 described

themselves as ultra-Orthodox but did not specify the community they were affiliated with; the rest were secular or religious but not ultra-Orthodox. For example, from the Home Front Command, three out of the four people interviewed identified themselves as ultra-Orthodox and one as secular. From the United Hatzalah Emergency Service, four out of the six people interviewed identified themselves as ultra-Orthodox. The response rate to the interviews was 75%. The interviews allowed us not only to gain new insights and obtain data from additional perspectives, but also to openly discuss the existing and proposed policy mechanisms and recommendations with key actors in a dialogic and interactive process.

In addition, focus groups were held with the rescue organization Magen David Adom, ultra-Orthodox educators and teachers, and an ultra-Orthodox organization for people with disabilities. Focus group is a moderated group discussion that allowed us to be attentive to the group interaction and the discursive dynamic between the participants (Barbour, 2010).

The interviews and focus groups were recorded, transcribed verbatim, and thematically analyzed both deductively and inductively according to key categories and themes. Pseudonyms are used for participants' names to maintain anonymity.

Finally, this socially engaged research (Golan et al., 2017; Orr, 2016-17) also aimed to promote practical positive change in the level of earthquake preparedness of the ultra-Orthodox community in Jerusalem. Based on an implementation of the research findings, we trained a class of ultra-Orthodox fourth year nursing students at the Jerusalem College of Technology in earthquake preparedness. These students trained numerous ultra-Orthodox families in Jerusalem using a culturally-adapted curriculum. Many of these family members agreed to become "ambassadors" in their community and train other community members using the localized curriculum. The people trained by the students filled out a short feedback questionnaire that allowed us to improve the training.

**4 Results**

**4.1 Findings from the Survey**

**4.1.1 Demographic Features**

The mean age of the respondents was 28, with an over-representation (63%) of men. Segmentation was carried out for each gender separately because of the different educational trajectories for men and women in the ultra-Orthodox education system. Most men graduated Yeshiva (an Orthodox Jewish seminary that focuses on the study of traditional religious texts), while most of the women had academic or nonacademic higher education. The majority (82%) lived in Jerusalem and its surrounding towns. The others lived in other ultra-Orthodox cities (Modi'in Illit, Bnei Brak, Elad) and in mixed cities. The sample represents the major ultra-Orthodox subgroups, including Hasidim, Lita'im, and Sephardim; and other subgroups such as "Olim" (first- or second-generation immigration) and "Baalei teshuva" (secular Jews who returned to practice Judaism). More than half had no smartphone or had a smartphone with limited accessibility to internet content for religious reasons (Table 2). Of those who did not own a smartphone, 69% answered the questionnaire via in-person interview. However, 31%

of the non-smartphone owners answered the online form; this implies that many ultra-Orthodox people have
(albeit limited) internet access. It should be noted that limited internet access was one of the considerations for
choosing in-person interviews for most of the respondents.[1]
**4.1.2 Earthquake Preparedness**
When asked about the likelihood of a disastrous earthquake occurring in Israel in the next five years, 55%
believed that the chances are nonexistent or low. The percentage of disbelief rose to 64% when respondents
were asked about the likelihood of an earthquake occurring in close proximity to their living area (Chart 1). To
the statement "I do what it takes to prepare for the possibility of an earthquake even if it costs money and
requires time," the majority (75%) responded negatively or said that they prepare minimally. Similarly, 81%
responded negatively to the statement "I actively look for information regarding earthquake preparedness."
Likewise, 77% of the respondents indicated that they did not take any preparedness actions such as stabilizing
bookshelves and keeping heavy objects close to the floor, and 52% said that they do not have the equipment
available for emergencies. The quantitative findings are presented in Appendix C.
One of the open-ended questions asked the respondents what they would do if an earthquake occurred. Many of
the respondents (40%), including those who replied that they did not know the rules of conduct, knew the basic
guideline of exiting to an open area. In response to another question, however, most of the respondents (68%)
said that they were not aware of the guideline of disconnecting electric and gas switches following an
earthquake. Fifty-nine percent of the respondents believed that they did not have the knowledge and aptitude
necessary to deal with an earthquake. Around half of the respondents believed that their home did not meet the
requirements of the law passed in Israel in 1995 setting a new standard regarding earthquake safety. Only 15%
of the respondents who are parents to small children discussed the rules of earthquake emergency with their
children. Three percent of these parents practiced the rules with their children. To the question, "Overall, have
you prepared for an upcoming earthquake in Israel?" only 6% answered that they have prepared or that they
have very much prepared. Men prepared more than women (p value = 0.013, $x^2$=10.639). Education, marital
status, community affiliation, and hometown did not influence preparation.
**4.1.3 Effect of Religion on Preparedness**
Regarding the religious views on earthquake preparedness, the respondents were asked in an open-ended
question if they believe that there is a Jewish religious obligation to prepare for disasters like earthquakes.
Approximately two-thirds (68%) answered positively, with some quoting specific commandments to prove their
point — for example, "but you shall greatly beware for your souls" (Deuteronomy 4:15), 7% stated that a
leading rabbi must rule on this matter, and 12% answered that they believe that there is no such obligation.
When asked more specifically if earthquake preparedness is a religious obligation enforced by the directive that
one must guard one's own life (hishtadlus), 46% strongly agreed (Chart 2).

---

[1]According to Cahaner and Malach (2019, p. 69), 49% of the ultra-Orthodox adults use the internet, in comparison to 89% among non-ultra-Orthodox Jews in Israel.

The respondents were asked whether they think that the fact that emergency preparedness guidelines are issued
by the national (largely secular) authorities has a negative effect on their community's preparedness level.
Twenty percent answered positively, saying that the source of the instructions has a negative effect. Another
20% believed that the source of the instructions has a positive effect on preparedness since it makes people take
the guidelines more seriously. Others believe that there is no existing connection between the source of the
instructions and preparedness level.
**4.1.4 Exposure to Earthquake Preparedness Publications**
About half (46%) of the study participants answered that they have been exposed to publications regarding
earthquake safety measures through various media such as newspapers, websites, radio, and direct mail.
Additional sources of exposure include hospitals, government offices, and schools. There was a significant
positive correlation between exposure to advertisements and preparing for the possibility of an earthquake in
Israel (p value = 0.002).
**4.1.5 Ways to Upgrade Preparedness**
We asked the respondents about what could convince or help the ultra-Orthodox public to prepare for an
earthquake. Twenty-five percent of the respondents answered that learning of the high probability of an
earthquake occurring in their area would convince them to prepare. Other factors they mentioned include having
a higher level of awareness, training, advertisements on the subject, the authority of a religious leader and other
state authorities, and educating the youth. When asked specifically if an instruction from a religious leader or a
ruling according to Jewish law would convince them to prepare for earthquakes, 68% of the respondents said
"yes." Thus, the four main ways that respondents identified as effective in raising awareness in the ultra-
Orthodox society are as follows: instruction from a religious leader, disseminating information in ways that are
adapted to the ultra-Orthodox society, community preparedness efforts, and youth education.
**4.2 Qualitative Findings**
A key issue discussed in the in-depth interviews and focus groups was how the religious faith and culture of
ultra-Orthodox Jews affect their preparedness. The findings can be divided into three main themes: the
characteristics of the ultra-Orthodox society that may have a negative effect during the preparation and
emergency phases; the characteristics of the ultra-Orthodox society that may have a positive effect during the
preparation and emergency phases; and strategies for improving the preparation and emergency phases. All the
information in this section is based on the qualitative results deducted from the interviews and focus groups with
relevant stakeholders.
**4.2.1 Characteristics That May Have a Negative Effect**
The Jewish ultra-Orthodox society has several characteristics that may have an adverse effect on its level of
preparedness for an earthquake and make functioning during emergencies difficult. The characteristics we will
discuss in this section include the following: certain aspects of the ultra-Orthodox worldview and guiding

philosophy, technological disparities, the insularity of educational institutions, attitudes toward state authorities, and low socioeconomic status.

**Worldview and Guiding Philosophy.** In the Jewish ultra-Orthodox society, various religious laws and viewpoints influence emergency preparedness. We identified two basic, ostensibly contradictory perspectives rooted in the Jewish tradition that lead to divergent approaches to emergency preparedness. One perspective can contribute to a religious preference not to prepare for earthquakes, while the other can lead to appropriate preparations. It is the deeply-held belief that everything that happens is God's will and preparation for a disaster cannot change God's decree that leads many ultra-Orthodox individuals to refrain from preparing for disasters. Likewise, some believe that the best way to prepare is through prayer, repentance, and studying sacred texts. Last, some consider preparedness efforts to be implying a lack of trust in God and, therefore, refrain from such activities. Some of these philosophies may make it difficult for the people to efficiently prepare for earthquakes. This point was brought out through the words of Shmuel, one of the leaders of an ultra-Orthodox emergency rescue organization: "During an emergency, we approached members of one ultra-Orthodox community asking for help. They immediately offered to pray, when in fact we were looking for help in rescue operations and evacuating the injured. There is totally a risk factor here that needs to be emphasized … Part of the ultra-Orthodox community believes that ultra-Orthodox faith and prayer would do more to protect than anything else."

Having said that, some of our interlocutors presented a more complex and nuanced perspective according to which their strong belief that everything that happens is God's will does not contradict the need to prepare for an earthquake. For example, Bilha, an ultra-Orthodox educator, explained: "We told the students that an earthquake is something that God does, and no one can do such a thing but God. If he does it, there is a reason why he does it." However, her take was that "It does not contradict the fact that we should learn to be careful and how to protect ourselves, and that the authorities should do their job."

**Technological Disparities.** Most interviewees spoke of a prevailing lack of awareness regarding earthquake risk and proper preparation. This may adversely affect functioning during emergencies and make it difficult for the rescue forces to operate. One main cause for this low awareness is the fact that the ultra-Orthodox public is less exposed to technology for religious reasons. Most people do not own a television at home, and many do not have access to the internet or radio. This impedes the preparation and emergency phases of disaster preparedness, since many ultra-Orthodox people are not exposed to information disseminated online. This challenge is outlined by Gershon, an officer at the Israel Defense Force's Home Front Command, which coordinates the government's response in emergency situations, especially regarding the protection of civilians: "Today, despite having the most advanced technology, it is not accessible to most of the ultra-Orthodox sector … Such communication with the ultra-Orthodox sector is limited. Some of them still use old forms of media such as street posters and leaflets."

**Insularity of Educational Institutions.** In Israel, the military-affiliated Home Front Command invests heavily in providing emergency training to children and youth through schools and other educational institutions (Home Front Command, 2020). We learned from our interviews with role holders from the Home Front Command and Ministry of Education that many ultra-Orthodox institutions do not allow

the government or military to enter educational institutions to deliver training even on crucial issues such as emergency preparedness. This is a decisive factor that leaves the ultra-Orthodox population less prepared for earthquakes. Gershon from the Home Front Command elaborated on this subject: "Everything related to the Israeli army, especially in our times, hold a lot of tension. It is complex. Very few institutions cooperate with us. The ultra-Orthodox education system usually does not allow military elements to enter." Though the Home Front Command has adapted its curriculum to the norms of the ultra-Orthodox society, most ultra-Orthodox schools still do not allow military personnel to conduct training. Therefore, the majority of the students do not receive any such instruction. Moreover, the Home Front Command has prepared booklets with safety information for students, adapted according to the ultra-Orthodox norms. However, according to Ya'ir, a safety officer at the Ministry of Education, many schools would not use these booklets because they are published by the Home Front Command.

**Attitude Toward State Authorities.** Opinions among the interviewees differed on the fact that earthquake preparedness guidelines come from the state and military authorities. Some of them insisted that members of the ultra-Orthodox population would not accept directives that come from the army or state institutions, while others believed that the ultra-Orthodox society would respectfully accept these instructions.

**Low Socioeconomic Status.** A few interviewees stated that the low socioeconomic status that characterizes the ultra-Orthodox society is another factor that may hamper earthquake preparedness. The interviewees reasoned that people immersed in daily survival spend less time thinking about preparing for future emergencies. Furthermore, people with limited means will find it difficult to buy and maintain essential emergency equipment.

### 4.2.2 Characteristics That May Have a Positive Effect

The Jewish ultra-Orthodox society features characteristics that may improve its level of preparedness and help its functioning in times of emergency. The interviewees talked about certain characteristics of the ultra-Orthodox community that may create a positive effect on the preparation and emergency phases of an earthquake. These characteristics include certain aspects of the ultra-Orthodox worldview and strong social capital.

**Worldview and Guiding Philosophy.** As mentioned earlier, there are two Jewish perspectives that result in opposing approaches to emergency preparedness. In the previous section, we explained a prominent view that has a negative effect on preparedness. Another set of Jewish laws, however, may actually have a positive effect. A large number of interviewees said that the ultra-Orthodox society is very strict about the religious precept to keep away from danger and threats. They pointed out that in Judaism, one may not rely on miracles to stay safe. Many of our interlocutors argued that the subject of preparedness would be acceptable if it were presented in such terms. Many research participants referred to specific Torah sources in their explanation of the duty to prepare for an earthquake. Akiva, an ultra-Orthodox educator, explained: "The Torah says: 'Be wholehearted

with the Lord, your God,[2] … a person should be concerned about the present and not be paranoid. However,
'You shall greatly beware for your souls.'[3] It requires us not to underestimate anything that can harm us, no
matter how close or far away it is. One should not panic; one should not be haunted by this thing. But, to prepare
for it … is part of the Halakhic requirement" (The Halakha is the collective body of Jewish religious laws). Rabi
Levy, a prominent rabbi that we interviewed, said that learning about earthquake preparedness can be
considered as Torah study (Jewish religious study) and, therefore, can be viewed as legitimate: "Caution is in
itself a learning that I would consider to be Torah study." The research participants proposed to emphasize and
leverage this religious point of view.
**Strong Social Capital.** The ultra-Orthodox community has a high level of social solidarity and sustains many
community organizations that provide medical and financial aid. The qualitative data reflect a general agreement
that one of the strengths of the ultra-Orthodox society from a disaster risk reduction point of view is its high
level of social involvement. A number of respondents noted that in the first few days after a potential
earthquake, they expect most of the aid to come from within the community and not from the authorities. For
this reason, a cohesive and engaged community, such as the ultra-Orthodox community, has a higher chance of
reducing harm in the emergency phase and thriving in the recovery phase. David, a senior official in an NGO
specializing in rescue and recovery, described how a large, connected ultra-Orthodox community can improvise
and manage during a disaster: "From the perspective of manpower, we have more organized and dynamic
manpower than any other place. There are between 70,000 and 130,000 Yeshiva students and young men. There
is a clear hierarchy. They can mobilize recruitment, know how to organize huge events, and know how to
translate that to improvisation and emergency assistance."
**4.2.3 Strategies to Improve the Preparation and Emergency Phases**
In this section, we will focus on those methods discussed by the participants that can help improve the ultra-
Orthodox population's earthquake preparedness and functioning during the emergency. The main ways to
improve these stages include the following: providing information, demands from state authorities, soliciting
support from religious leaders or community figures, school training, family activities, advertising, adapting the
state's training to the ultra-Orthodox public, empowering the ultra-Orthodox organizations in leading
preparedness efforts, leveraging solidarity, and adapting technologies.
**Providing Information.** Many interviewees agreed that it is essential to invest in explaining to the public "what
an earthquake is." Various ways were suggested to publicize the information, including children books,
community leaders, workshops for parents, conferences, and existing community-based organizations. Several
respondents said that this process should emphasize the importance of moving to an open area during an
earthquake.
Most of our interlocutors were knowledgeable about the earthquake risk because of their professional role.
However, those who did not think that the risk is real demonstrated the need for providing and disseminating

---

[2]Deuteronomy 18:13
[3]Deuteronomy 4:15

information. Thus, when we asked the ultra-Orthodox teachers whether a ruling from rabbis would help improve preparedness, they said that it would help, but is unnecessary because Halakhic rulings deal with more important issues: "It is not one of the matters of utmost importance that currently preoccupies the ultra-Orthodox public … If there is a ruling, then it will probably have a greater impact; but, our rulings focus on more relevant issues now." Another teacher supported her: "Come on, in the Land of Israel, we hardly suffer from earthquakes. So, there is nothing much to worry; we are not like Japan." The precondition for applying effective methods like rabbis' ruling is increasing the public understanding of the risk.

Moreover, rabbis and community leaders felt that they themselves do not know enough about the subject and were willing to learn more to be able to take action. Rabbi Cohen told us, "I know very little. I know about earthquakes and about the fear. But, I do not know of any practical thing that I can advise people to do. Just talking to them will only create panic. I have nothing to do; I cannot do anything." Rabbi Levy said: "The last time there was an earthquake, I remember, people did not escape from the buildings. There was an earthquake. People felt it. I thought that some construction work was happening in the synagogue where I studied. It was interrupting my studies. But, I did not think of an earthquake. People are not aware of the possibility of a disaster." These and other rabbis and community leaders were open to learn about the practicalities of earthquake preparedness as the first step in their engagement in promoting the religious community's preparedness.

**Demands from State Authorities.** Participants in the focus groups with educators in the ultra-Orthodox institutions described a bad and worrying situation in terms of knowledge, preparedness, and drill in the ultra-Orthodox education system. They attributed this problematic situation to the very few binding requirements on the part of the government and municipality. Rivka said: "I think that if there was a demand [from the authorities], we would fulfill it. It does not seem something out of this world. But, we do not conduct drills as the authorities do not require it. We are incredibly busy and we have plenty of other things to do. No one demands, no one comes to see, no one asks." In response, Akiva added that if there is no formal requirement to prepare a lesson on this subject, he will not spend time working on it. He also contended that the state should provide appropriate pedagogical materials to the ultra-Orthodox institutions. These research participants believed that a formal, binding, measurable, and unequivocal requirement from state authorities would be helpful in placing this issue high in their institutional agenda.

**Soliciting Support.** The ultra-Orthodox society places great value on obeying religious authorities. Interviewees often believed that efforts to disseminate earthquake preparedness guidelines to the public must be supported by the ultra-Orthodox religious authorities. Apart from receiving their support on earthquake preparedness training, interviewees identified other critical matters where they can help. For example, Gershon from the Home Front Command said that there is an existing technology that can send a text message to "call only" phones (audio devices without texting options) and issue alerts about an earthquake in progress. Many ultra-Orthodox Jews do not allow the use of text messaging for religious reasons, and use "call only" phones. Therefore, implementing this technology would require the support of religious leaders.

There were differences of opinion among the interviewees regarding which religious authority figures should be approached. Some said that support must come from the accepted and esteemed rabbis, while others believed

that it is enough to receive support from the community leaders, directors of major educational institutions, or public activists. However, the rabbis and community leaders agreed that in order to be effective, the message should come from both the rabbis and experts in the earthquake field by working closely together. Rabbi Cohen explained: "If an expert would come and talk, I do not know how many people would come to hear him … I think that a combination of an expert and a rabbi, who will say 'listen, I saw the data, I am telling you that we must prepare,' will work in religious communities."

**School Trainings.** Respondents frequently said that an effective way to improve the preparedness of the ultra-Orthodox population is through lectures, activities, and seminars in educational institutions. They surmised that children and adolescents who undergo these sessions could raise the awareness of their entire family. Mendy, a safety officer in an ultra-Orthodox city, explained the importance of school training: "We invest a lot in instruction and preparation ahead of time. We do it through the schools because it is very difficult for us to gather the population, sit them down, and give lectures and sermons. We think that the ones we trained twenty years ago are already parents today … We invest in lectures, study days, instructions, and all that we can to give them the information."

**Family Activities.** The ultra-Orthodox society is very family-centered. Therefore, family ties can be a means of raising awareness regarding earthquake preparedness. It is reasonable to assume that parents who undergo training will pass on the information to their children. Shmuel, one of the leaders of an ultra-Orthodox emergency rescue organization, recommended conducting large events for ultra-Orthodox parents and children that will include activities such as earthquake training, demonstration of rescue techniques, emergency kits, publishing vital emergency preparedness documents, and holding lectures for parents and ultra-Orthodox public figures. Research participants emphasized the impact of interactive, participatory, and experiential activities that are more enjoyable, memorable, and effective.

**Advertising.** Another effective way to raise awareness is to publicize the issue among the ultra-Orthodox population using appropriate media. These include publishing the guidelines in ultra-Orthodox newspapers, and distributing street ads and neighborhood leaflets for free to every household. Further, the guidelines should also be published in ultra-Orthodox news websites and radio channels.

**Adapting the State's Trainings.** A considerable part of the ultra-Orthodox population is reluctant to accept directives issued by the state and military authorities. For this reason, the interviewees recommend removing all national government symbols from the earthquake preparedness literature. Moreover, respondents repeatedly recommended that the soldiers who deliver earthquake preparedness trainings in schools should not wear uniforms. Ya'ir, a safety officer at the ultra-Orthodox education system, explained the significance of the way the instructors introduce themselves: "The way he introduces himself to the ultra-Orthodox education system is crucial. He should introduce himself as a representative who has come to help and not as a representative of the Home Front Command … He must say that it is a religious obligation to prepare."

Using religiously and culturally adapted and appropriate language and concepts in trainings is crucial. For instance, when we had trained the ultra-Orthodox students who later trained other ultra-Orthodox community members, one of our slides that discussed the DSFS included the words "in the past 10–20 million years." The

students asked us to modify these words because the ultra-Orthodox people strictly believe that the world was created by God only 5,780 years ago, as written in the Torah. According to the students, the original slide would immediately alienate the ultra-Orthodox audience who would simply stop listening as the training contradicts their religious beliefs. This ostensibly minor detail, which can easily be fixed by using a broader term like "for many years," demonstrates the need to carefully adapt and localize the trainings in close collaboration with the ultra-Orthodox actors. The ultra-Orthodox students, who are familiar with both the ultra-Orthodox worldview and academic discourse, can serve as intermediaries and translators between these worlds.

**Empowering Organizations in Leading Efforts**. Several respondents noted that one of the ways to improve the population's preparedness for an earthquake is by transferring the entire issue from the Home Front Command to a civilian body that is considered legitimate by the ultra-Orthodox society. For instance, the government can conduct community-based trainings in ultra-Orthodox schools with ultra-Orthodox educators rather than soldiers as trainers.

**Leveraging Solidarity.** Research participants suggested practical ways to leverage the high level of social solidarity, mutual support, and strong social networks in the ultra-Orthodox communities. For instance, Yosef worked for an ultra-Orthodox rescue and recovery organization and volunteered at a community residential home for people with difficult physical disabilities in an ultra-Orthodox neighborhood in Jerusalem. He suggested training people in this neighborhood so that they can come and help the residential staff when an earthquake occurs, especially in evacuating residents with disabilities from the upper floors when the elevator does not function. This is essential due to the small number of staff members (especially at night) and because the evacuation of these residents is very complex. According to Yosef, these neighbors will be able to provide emergency assistance long before the professional rescuers arrive and can, thus, save lives. This project requires strong community engagement, commitment, and dedication that indeed exist in the ultra-Orthodox neighborhoods.

The strong solidarity in the ultra-Orthodox society is also reflected in an extensive and well-organized donation system (food donations, for instance). Some research participants proposed to include the necessary emergency equipment, such as canned food and flashlights, in this donation system. This will help to cope with the difficulties that derive from the high rate of poverty in the ultra-Orthodox society and also make earthquake preparedness part of the day-to-day discourse, thus raising awareness on this issue.

**Adapting Technologies.** Many respondents believed that preparedness for the emergency stage can be improved using various technologies adapted to the ultra-Orthodox population's culture. One idea is the distribution of single-channel radios that would only be used during emergencies to receive guidance from the authorities. Although many leading ultra-Orthodox rabbis do not allow the use of radio, there is hope that this idea would get acceptance as the device has access to only one emergency channel.

## 5 Discussion

It is worthwhile to compare the level of preparedness of the Jewish ultra-Orthodox population in Israel to that of the general Israeli public. Ya'ar et al. (2015) explored the Israeli population's perception of the occurrence of a

strong earthquake in their country, and found that 33% of the Israelis do not believe that a devastating earthquake will occur in Israel in the near future. The level of disbelief rose to 45% when asked about the occurrence of a devastating earthquake in their close proximity in the near future. In comparison, our ultra-Orthodox respondents showed a higher level of disbelief regarding the occurrence of a devastating earthquake in Israel in the near future (55%) and in their close proximity in the near future (64%). These differences are also reflected in percentages of the people who agree with statements regarding knowledge and readiness to prepare for an earthquake (Chart 3).

We found that religious beliefs and worldviews can have both a positive and negative impact on disaster preparedness. Most of the respondents (75%) said that they have not prepared much for such a disaster. This finding aligns with previous studies on minority groups' low level of self-protection, preparedness, and hazard knowledge in comparison to the majority group (e.g., Lucini, 2014; Maldonado et al., 2016). Specifically, half of our survey respondents indicated that the buildings they live in do not meet the legal safety standards despite the fact that some of their neighborhoods are in areas with increased ground shaking hazard during an earthquake (Salamon et al., 2010) (Figure 2). Alarmingly, none of our respondents recommended retrofitting, a crucial strategy for minimizing the harm caused by earthquakes (Bertero and Bozorgnia, 2004). Our interviews with relevant stakeholders further confirmed that very few ultra-Orthodox people are interested in retrofitting. They explained that this is because of a lack of awareness regarding three factors: the potential impact of earthquakes, significance of building conditions in reducing damage and casualties, and government's willingness to support retrofitting. Other reasons include the intangibility of the danger and low economic status of the ultra-Orthodox society.

On the positive side, most respondents believed that there is a religious obligation to prepare for a disaster, stating that the Jewish religion forbids one to rely on miracles and one must do all they possibly can to stay safe. This finding stands in opposition to many studies that have found that religion often creates a fatalistic attitude that hinders disaster preparedness (Baytiyeh and Naja, 2014; Plapp and Werner, 2006; Sun et al., 2018; Yari et al., 2019). Uniquely, Gianisa and Le De (2018) support our finding; they described how religious beliefs urge preparation among Muslims. In our qualitative research, however, many interviewees said that fatalistic attitudes are common in the ultra-Orthodox society, negatively affecting their state of preparedness, as described by other scholars regarding different religious groups. These interviewees said, for example, that some ultra-Orthodox people believe that one should pray to avoid a disaster instead of actively preparing for it. The more complex and subtle approach reflected in the aforementioned words of Bilha maintains that the strong belief that everything that happens is God's will does not contradict the need to prepare for an earthquake. Disaster risk reduction efforts can include illumination and reinforcement of this complex position, for example, in schools, like Bilha does. This strategy is in keeping with Azim and Islam's (2016) proposal to incorporate certain interpretations of religious sources to improve preparedness (cf. Baytiyeh and Naja's, 2014).

We found that lack of exposure to the mainstream media lowers the ultra-Orthodox society's preparedness level. This finding is in line with previous studies (Kellman, 2011; Ya'ar et al., 2015). We also found that in some communities within the ultra-Orthodox society, any guidelines that come from national authorities are viewed with suspicion. This attribute can be troubling when adherence to these messages is a matter of health and safety

(Spence et al., 2007). Many interviewees described that the ultra-Orthodox education system is characterized by
insularity, forbidding entry to military personnel to deliver emergency training and sometimes disallowing the
distribution of information pamphlets written by state officials. This finding is bothering considering the key
role of education in improving preparedness in many other countries (Baytiyeh, 2017; Lucini, 2014; Smawfield,
2013; Yari et al., 2019). However, some participants insisted that the state authorities should define clear and
binding requirements for the ultra-Orthodox educational institutions and provide them with appropriate
pedagogical materials. Hence, a delicate balance seems to be needed between the ultra-Orthodox community
autonomy, on the one hand, and supervision and a strict professional standard defined by state authorities, on the
other hand. For example, the state can define a clear standard of preparedness, but allow community-based
organizations the flexibility and freedom to choose how to reach the shared goals with the support of state funds
and resources. The state will monitor and examine the local organizations' achievements in fulfilling these goals.
It is essential to build trust and improve the mutual communication channels and cooperation between the
relevant state institutions, and ultra-Orthodox rabbis and community leaders. The lack of such trust and
communication has had extremely negative ramifications during the COVID-19 pandemic that hit the ultra-
Orthodox community much harder than the other communities in Israel (Waitzberg et al., 2020). Both sides
must engage in a respectful dialogue, acknowledging the knowledge, expertise, capabilities, beliefs, needs, and
difficulties of the other side. They should recognize that they both share the responsibility for the current low
level of hazard knowledge. They must also share the responsibility to fundamentally change this bothersome
situation.
Many of our interviewees predicted a relatively successful recovery in the ultra-Orthodox society following an
earthquake. This is due to the strong social capital and resilience, which distinguish the ultra-Orthodox
community. The effective community-based ultra-Orthodox religious institutions can play a pivotal role in the
reconstruction and rehabilitation phase. Our research results support the finding of previous studies that
established social networks are a strong predictor of population recovery after an earthquake (Aldrich, 2011;
Gianisa and Le De, 2018). Community support, social bonding, and shared beliefs are especially helpful for
religious minorities following a disaster (Ngin et al., 2020). However, scholars have demonstrated that political
controversies and various social-cultural divides may constitute an obstacle to recovery (Cheema et al., 2014;
Ngin et al., 2020). Therefore, building trust, ongoing dialogical communication, and long-term collaboration
between the ultra-Orthodox community and state authorities are crucial in the recovery, preparation, and
emergency phases. Furthermore, the prevailing poverty in the ultra-Orthodox society is another considerable
obstacle to recovery and preparation (cf. Shapira et al., 2018).
We asked the research participants' opinion on how to effectively improve preparedness in the Jewish ultra-
Orthodox society. Most respondents answered that the key to improving disaster management in the ultra-
Orthodox community is to raise earthquake hazard awareness. In line with Appleby-Arnold et al. (2018), the
respondents thought that national-level policymakers must consider all the characteristics of the ultra-Orthodox
society that we discussed, before planning disaster management strategies. All tactics should be adjusted and
adapted according to the ultra-Orthodox norms. The participants believed that guidelines on earthquake
explained and approved collaboratively by both the rabbis and experts will be very effective. This finding is
supported by recent data during the COVID-19 pandemic that shows that the ultra-Orthodox people tend to
comply with prominent ultra-Orthodox rabbis as well as expert physicians (Kalagy et al., 2020).
Finally, we would like to discuss the limitations of the research methods used. In the survey, we used non-
probability sampling that may not represent the entire ultra-Orthodox population in Israel. There were only 288
respondents, and there was an over-representation of men. The average age was young, and a great majority of
respondents lived in Jerusalem and its surrounding areas. However, in our opinion, our sample also has
considerable advantages. First, the in-person interview survey was conducted by ultra-Orthodox or Orthodox
surveyors, enabling us to reach out to ultra-Orthodox people who would probably refuse to respond to a
telephone survey from an academic institution. A telephone survey, which is faster and simpler, has a lower
response rate and might involve considerable bias when it comes to the ultra-Orthodox public.[4] Second, the
survey conducted in face-to-face interviews allowed a very long and detailed questionnaire to be answered. It
allowed the elaboration of questions and answers, ensured that the questions and answers were understood, and
avoided offhand answers. None of this happens in a telephone or internet survey. Third, focusing on particular
communities, such as the ultra-Orthodox communities in Jerusalem and its surrounding areas, allows a deeper
understanding of their needs and challenges and enables the possibility of working with them over time. For
example, it allows efficient training in these communities, like the training led by dozens of our ultra-Orthodox
students in several ultra-Orthodox communities in Jerusalem, after we trained them. Last, the young people
interviewed either had or were about to have children. We have learned from the literature that children's
education is one of the most impactful tools in disaster risk reduction. Hence, we wanted to work with these
young people.
**6 Conclusion**
Our research questions included three main themes: the actual state of earthquake preparedness in the ultra-
Orthodox society, characteristics that may hinder or promote preparedness, and ways of improving
preparedness. The findings from the first two questions are summarized in a SWOT (Strengths, Weaknesses,
Opportunities, Threats) analysis that identifies the areas where the community is prepared and where it lacks
preparedness. Furthermore, it identifies the community characteristics that hinder and increase preparedness
(Table 3). To answer the third research question, we asked the research participants to share their opinions on
venues for increasing preparedness. These venues include the following:
• Providing information and raising awareness of the ultra-Orthodox community and its religious leaders.
• The state should set clear, binding, and measurable preparedness standards and goals but allow the local
communities the autonomy and flexibility to decide how to reach these goals with state funds and
monitoring.

---

[4]Before the current study, surveyors refrained from using in-person interviews in this subject. This resulted in a
lack of sufficient knowledge about the ultra-Orthodox society's preparedness. A member of the National
Steering Committee for Earthquake Preparedness shared in an interview: "From our experience, the ultra-
Orthodox society is less open to cooperate in surveys … We do not know their level of preparedness."

- Collaborating with civil organizations that are perceived as legitimate by the ultra-Orthodox society that can lead and coordinate the preparation initiatives in this society.

- Receiving the support of ultra-Orthodox religious authorities for any project or campaign.

- Close collaboration between rabbis and experts on earthquakes in reaching out to the ultra-Orthodox public.

- Lectures, activities, drills, and seminars in ultra-Orthodox educational institutions and during family activities.

- Publishing the guidelines in ultra-Orthodox newspapers, street ads, neighborhood leaflets, news websites, and radio channels.

- Removing any national and government symbols from earthquake preparedness publications for ultra-Orthodox communities. Preferably, trainers should be ultra-Orthodox community members; if soldiers deliver earthquake preparedness trainings in schools, they should not wear uniforms.

- Leveraging the ultra-Orthodox community's strong social capital during preparedness and emergency phases.

- Adapting emergency technologies to make them acceptable for the community.

The findings establish that religion is a significant factor that influences all stages of disaster response. Therefore, it must be taken into consideration when attempting to improve earthquake preparedness. This research highlights some of the features of a religious minority group that may affect its preparedness, whether positively or negatively, and suggests avenues for improving its level of preparedness. While previous studies have examined the impact of religion on the level of preparedness in general, our findings add significant knowledge to the existing literature on the influence of religion on preparing for a natural disaster among minority groups, especially when most state officials and policymakers are part of the secular majority. The findings of this study can be generalized and used by policymakers worldwide when attempting to improve disaster management of any religious group.

**7 Recommendations**

We would like to conclude with several recommendations based on this study that can possibly improve disaster management among religious, sociocultural, and other minority groups worldwide:

- In-depth study of every social-cultural and religious group, and the required adaptations that will upgrade their earthquake preparedness, building on their local knowledge.

- Before approaching a particular group, it is helpful to receive support from its leaders.

- Establishment of a formal mechanism for reciprocal and continuing dialogue and collaboration between the relevant state institutions, and community leaders and representatives. For this purpose,

both parties should acknowledge the necessity to build trust and joint work networks to improve the
level of preparedness.
• Emergency preparedness representatives must be acceptable for the community; their dress code and
language must be appropriate; and their instructions must be adapted to the community's needs and
social, cultural, and religious characteristics.
• The preparation guidelines should be spread on tracks to which the group is exposed. In sectors
unexposed to media for cultural or religious reasons, guidelines should be published in alternative
ways.
• Another venue for publicizing earthquake preparation guidelines is in hardware stores where cheap
rescue kits for the family and home security items can be sold.
• Bolstering existing local resources and community organizations by giving them official
responsibilities in the area of earthquake preparedness, since these organizations are acceptable for the
community.
• Government offices should proactively raise awareness regarding the importance of retrofitting. The
information about the financial support available to citizens who want to retrofit their houses should be
more accessible.
• The government should suggest practical emergency skills considering the population density and form
of construction that characterize the specific social group.

## Data availability

The data is unavailable for review as we promised complete confidentiality and anonymity to the research participants adhering to the ethical principles of social science research.

## Team list

Zvika Orr[1], Tehila Erblich[1], Shifra Gottlieb[1], Osnat Barnea[2], Moshe Weinstein[3], Amotz Agnon[2]

[1]Department of Nursing, Jerusalem College of Technology, Jerusalem, 9116001, Israel
[2]Neev Center for Geoinfomatics, Fredy & Nadine Herrmann Institute of Earth Sciences, The Hebrew University of Jerusalem, Jerusalem, 9190401, Israel
[3]Department of Electro-Optics Engineering, Jerusalem College of Technology, Jerusalem, 9116001, Israel

## Author contribution

ZO and AA designed and led the study, developed the methodology, and contributed to data interpretation. TE conducted interviews and analyzed the qualitative data. SG created the questionnaire and performed the statistical analysis. OB contributed to the questionnaire and data collection. MW initiated the study and conducted interviews. All authors contributed to the article preparation.

## Competing interests

The authors declare that they have no conflict of interest.

## Financial support

This research was supported by the Ministry of Science and Technology, Israel.

## Acknowledgements

We gratefully acknowledge the funding of the Ministry of Science and Technology, Israel. We thank the interviewees for participating in the study. We are also grateful to the dedicated seminar students and research assistants at the Jerusalem College of Technology for their help.

**Chart 1 – Level of belief that a disastrous earthquake will occur in Israel/in your area in the next five**
**years**
                Level of belief that a disastrous earthquake will occur in Israel/in your area in the
next five years

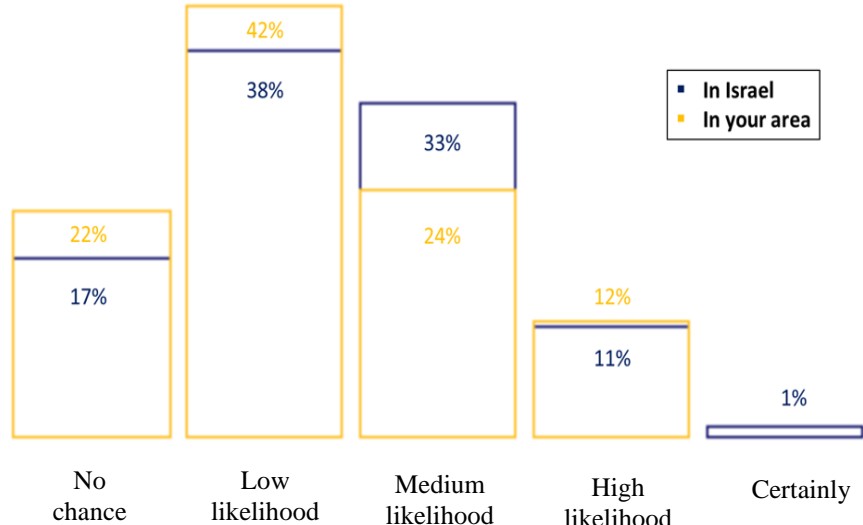

**Chart 2 – Agreement with the statement that earthquake preparedness is a religious obligation enforced**

**by the directive that one must guard one's own life (hishtadlus)**


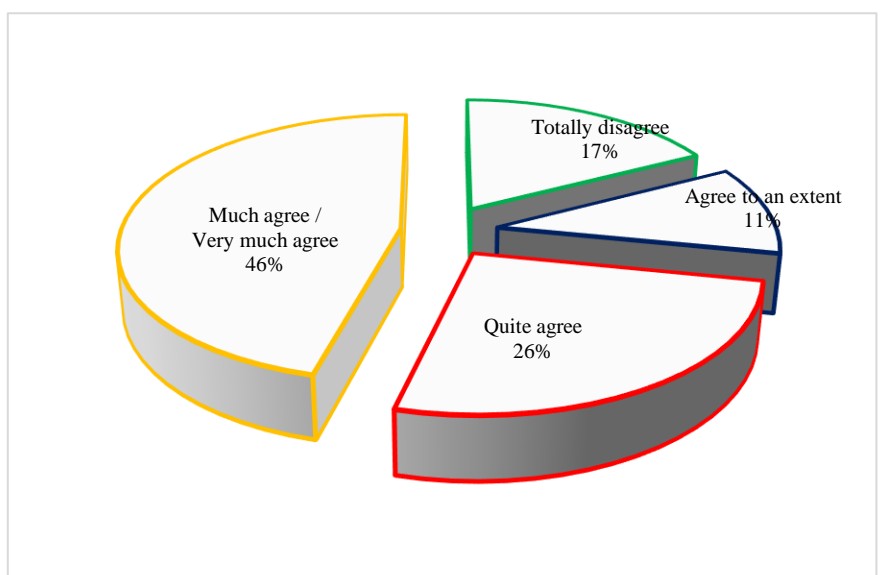



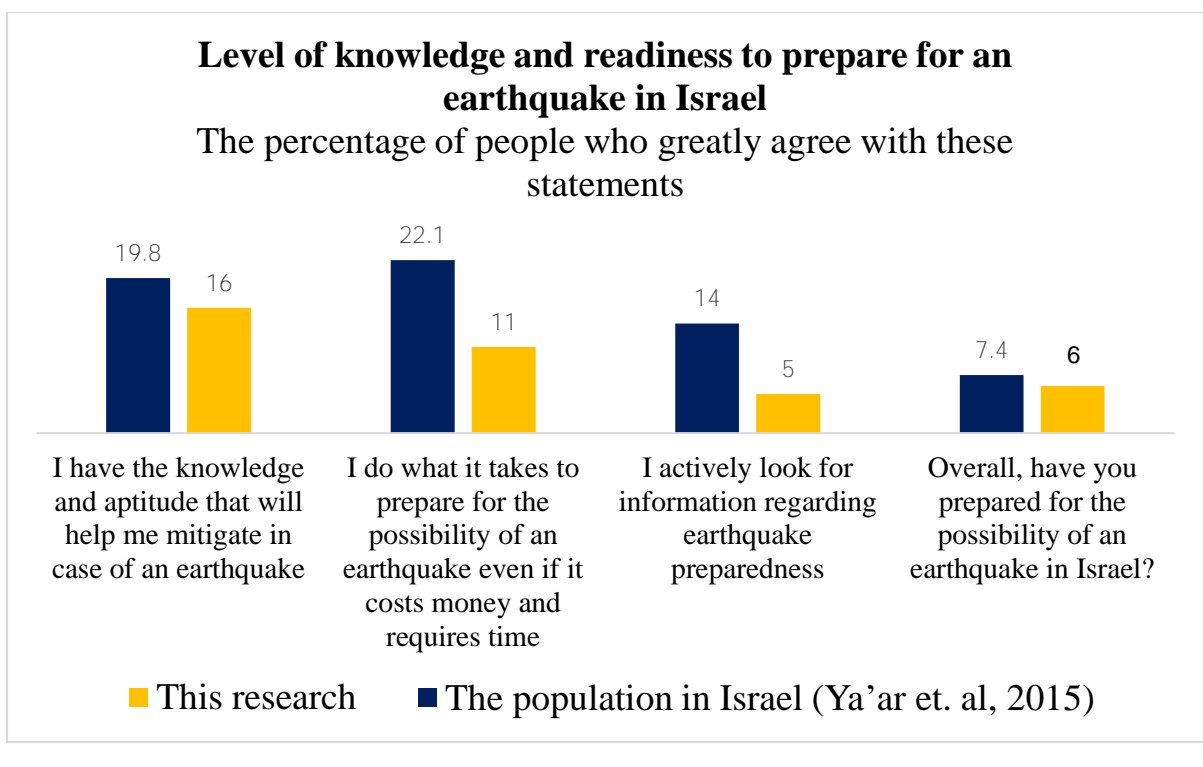


 **Table 1 – Listing of historic (pre-instrumental) earthquakes that damaged Jerusalem since Roman times**

| Year CE | Estimated magnitude | Jerusalem intensity [a] | Comments |
|---|---|---|---|
| 363 | 6-6.5 | VIII | Likely source: the Carmel branch |
| 418/419 | 6-6.5 | ≥VI | Likely source: Jordan Valley |
| 634/635 | ~6 | ~VII | Northern Wadi Araba |
| 659/660 | ~6 | VI-VII [b] | Jordan Valley |
| 746 to 757 | 6.5-7.5 | VIII-IX | Multiple rupture along rift |
| (1016) | ? | VII | Local event, dubious |
| 1033 | 6.0 | VIII-IX | Holy sites damaged [c] |
| 1068 | 6.0 | VII | Dome of the Rock (or the rock) said to crack [c] |
| 1105 | n/a | V | Panic, no damage |
| 1113 | n/a | V | Panic, no damage |
| 1117 | n/a | conflicting accounts | Located to the Lebanese coast [c] or Jerusalem [d] |
| 1293 | 6.5-7 | No reports | Likely between VIII (Ramla) and IX (Karak) |
| 1458 | 6.5-7 | ~VII | Holy Sepulchre destroyed; a minaret collapsed |
| 1504 | n/a | V | Three shocks, no damage |
| 1546 | 6-6.5 | VIII | *"Damage to all tall houses"; "fell... Al Aqsa"* [e] |
| 1557 | n/a | VII | Several buildings collapsed [f] |
| 1643 | n/a | III-IV | Reported panic [f] |
| 1753 | n/a | III-VI | Felt in Jerusalem [f] |
| 1817 | n/a | VI-VII | Two churches seriously damaged [f] |
| 1834 | 6-6.5 | VI-VII | Several churches damaged, few collapsed [f] |
| 1843-1857 | n/a | IV-V | Several shocks felt [f] |
| 1859 | n/a | IV-VI | A strong earthquake felt [f] |
| 1863-1873 | n/a | III-V | Four to five shocks felt [f] |
| 1874 | n/a | IV-VI | A strong earthquake felt [f] |
| 1877 | n/a | III-V | Two earthquakes felt (15 February; 14 March) [f] |

| 1879 | n/a | III-V | A shock felt [f] |
|------|-----|-------|------------------|
| 1885-1889 | n/a | III-IV | Two slight shocks felt [f] |
| 1893 | n/a | III-V | An earthquake felt [f] |


[a] Intensity: local level of damage; The table below lists a key for reading the numerals. [b] Calculated by Langgut
et al. (2016).
[c] Guidoboni & Comastri, 2005. [d] Ambraseys (2009, p. 291), from "Historia Hierosolymitana of about 1122."
[e] Ambraseys (2009), p. 445.  [f] From catalogues of Amiran et al. (1994) and/or Ambraseys (2009).

| *Intensity* | III | IV | V | VI | VII | VIII | IX |
|-------------|-----|-----|---|----|-----|------|-----|
| *Level* | light | moderate | relatively strong | strong | very strong | destructive | ruinous |
| *Effects* | short shock, a few may realize that the earth quakes | furniture tremble, felt by a few outdoors, indoors feels like ship over a rough sea | freely hung objects swing, thin branches sway, liquids spill, waking of the sleeping, occasional panic | panic, liquids shake, small damage to solid houses, widespread panic, cracks in plaster | moderate damage to numerous solidly built buildings, like small fissures in walls | heavy destructions to about one fourth of the houses; some collapse | about half of the stone houses heavily destroyed, moat become uninhabitable |


MCS - Mercali-Cancani-Sieberg intensity scale (only seven out of twelve levels shown). Abstracted
and rephrased from Ferrari & Guidoboni, 2000.

**Table 2 – Demographic features of the sample**

| | | %, unless otherwise stated |
|---|---|---|
| **Age** | mean | 28 years |
| | standard deviation | 10.22 years |
| | range | 18–67 years |
| **Gender** | men | 63 |
| | women | 37 |
| **Marital status** | married | 59 |
| | single | 39 |
| | divorced / widower | 2 |
| **Children** | have children | 50 |
| | mean number of children | 3.7 children |
| | range | 1-12 children |
| **Economic status** | below average | 23 |
| | average | 53 |
| | above average | 16 |
| | refused to comment | 8 |
| **Education - men** | non ultra-Orthodox high school education | 7 |
| | nonacademic higher education | 5 |
| | academic education | 20 |
| | Yeshiva | 68 |
| **Education - women** | primary education | 5 |
| | high school education | 9 |
| | nonacademic higher education | 27 |
| | academic education | 59 |
| **Living area** | Jerusalem and surroundings | 82 |
| **Ultra-Orthodox subgroup** | Hasidim | 18 |
| | Lita'im (Lithuanian Jews) | 28 |
| | Sephardim | 25 |
| | Other (e.g., Olim, Baalei teshuva) | 29 |
| **Mobile phone type** | kosher phone (no SMS, no internet) | 29 |
| | basic phone / protected smartphone (SMS, limited internet) | 24 |
| | unprotected smartphone (internet) | 47 |


     **Table 3 – SWOT (Strengths, Weaknesses, Opportunities, Threats) analysis summarizing the state of**

     **earthquake preparedness in the ultra-Orthodox society and characteristics that may hinder or promote it**


| | Strengths | Weaknesses |
|---|---|---|
| Actual State of Preparedness | Many (40%) know the basic earthquake emergency guideline of exiting to an open area.<br><br>Almost half of the respondents were exposed to advertisements on the subject. | The majority do not believe that a devastating earthquake will occur in their area in the near future.<br><br>The majority have not made the necessary preparations.<br><br>Many are unfamiliar with the emergency response guidelines.<br><br>Over half of the respondents have not stocked on equipment and supplies necessary for emergencies.<br><br>Half of the homes are not built according to the legal standard.<br><br>Very few discussed the subject with their children or practiced the rules with them. |

| | Opportunities | Threats |
|---|---|---|
| Opportunities and Threats for Improving Preparedness | The majority believe that there is a religious obligation to prepare for a potential disaster.<br><br>The community has a strong social capital, which can be an advantage in the preparation, emergency, and restoration phases.<br><br>Preparedness perspectives based on strong social capital can be expressed in several areas:<br><br>• Obedience to religious leaders – by issuing a directive for preparation.<br><br>• NGOs – Provide simple and inexpensive home security measures; issue detailed instructions for preparation in appropriate language, content, and distribution; organization of trainings in coordination with the authorities and experts.<br><br>• Concern for each other and solidarity – Sharing tips for strengthening structures and preparation of emergency supplies; assistance in buying equipment.<br><br>• Train neighbors to come and help vulnerable people in their area when an earthquake occurs.<br><br>• Entrepreneurship – Initiatives to strengthen buildings, consulting services on the subject, hardware stores where basic home security items can be sold. | Religious belief that disasters are God's will.<br><br>Belief that the appropriate means of preparation is through prayer and not through action.<br><br>Low exposure to information and the media.<br><br>Difficulty in using educational institutions as a tool.<br><br>Suspicion towards instructions brought by state authorities.<br><br>Low socioeconomic status.<br><br>Minimal demands from state authorities to educational institutions. |

**Figure 1 – (a) Estimated spatial extent of ruptures from historic periods along the DSFS. (b) The DSFS main branches**

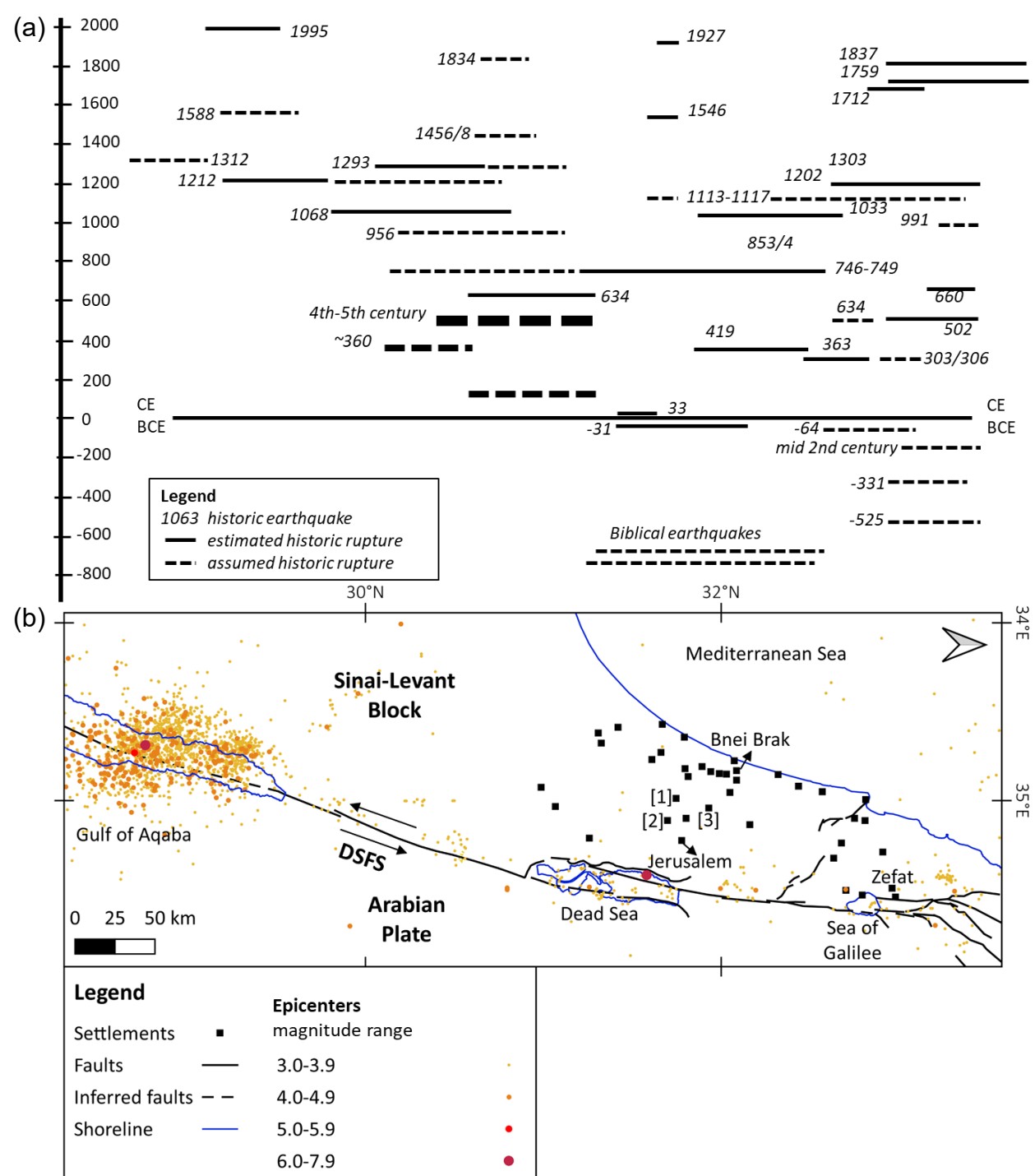

Figure 1: (a) - Estimated spatial extent of ruptures from historic periods along the DSFS (after Agnon, 2014). The position of the events in the figure is projected from the map of the DSFS below in (b).
(b) - The DSFS main branches (modified after Hamiel et al., 2018; Hofstteter et al., 1996; Kagan et al., 2011; Politi, 2011; Sharon et al., 2020) over epicenters recorded (since 1900) with Md>3 (www.gii.co.il).
Black solid squares mark settlements with prominent ultra-Orthodox populations (over a thousand ultra-Orthodox residents according to Shahak, 2017); the largest population is in Jerusalem and Bnei Brak.
Secondary yet prominent are Beit Shemesh [1], Beitar Illit [2], and Modi'in Illit [3].

**Figure 2 – Map of ultra-Orthodox neighborhoods in Jerusalem**

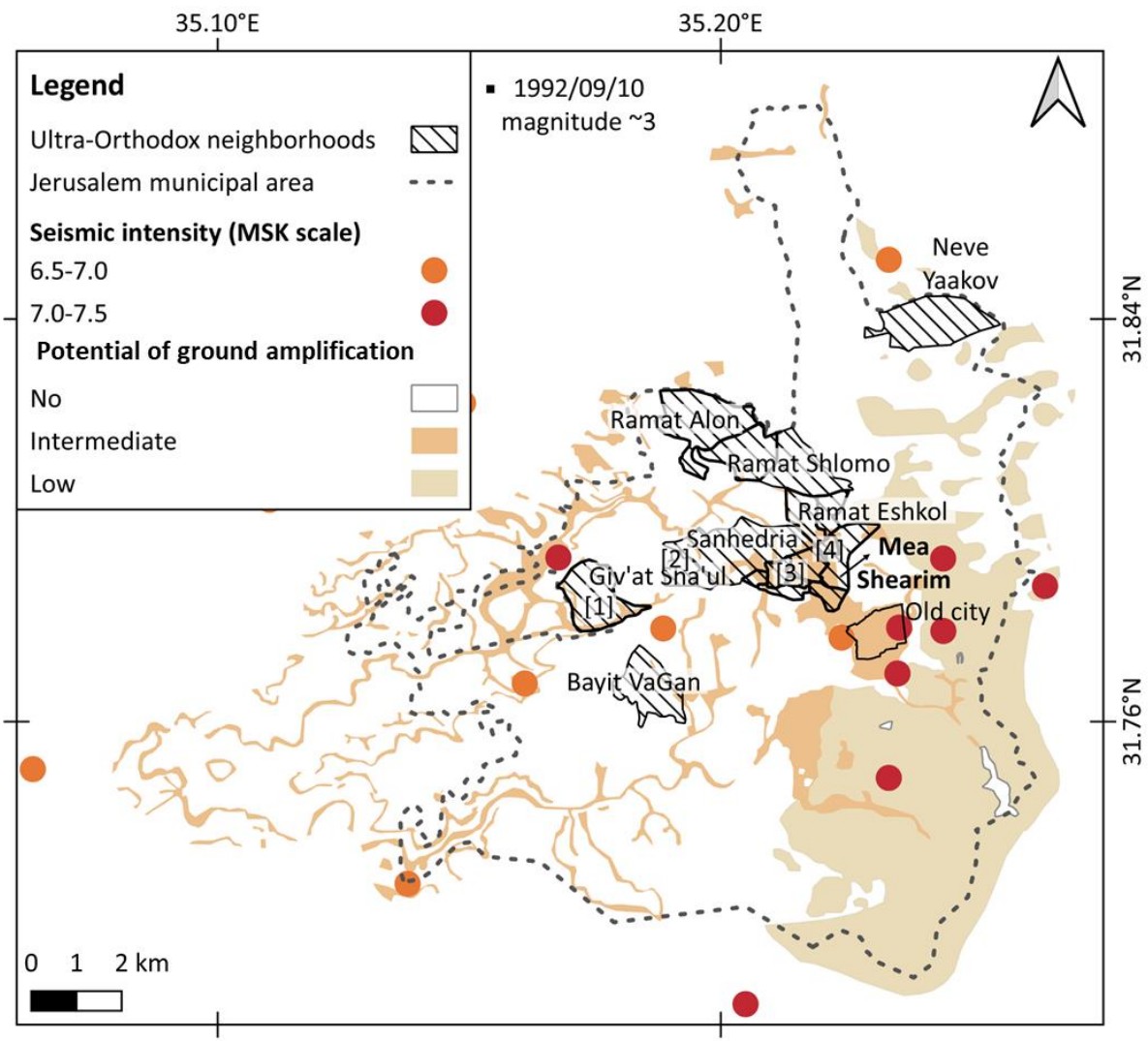

Figure 2: Map of ultra-Orthodox neighborhoods in Jerusalem (modified after Korach and Choshen, 2020; Golan-Agnon, 2020).
An ultra-Orthodox neighborhood is defined as a neighborhood where over 50% of the Jewish population aged 15 or older are ultra-Orthodox (Korach and Choshen, 2020, p. 25).
The background map outlines soil with a tendency to amplify shaking (Salamon et al., 2010). This is based on soil and rock classification and needs verification.
[1] – Har Nof, [2] – Romema, [3] – Ge'ula, [4] – Shmuel HaNavi.
The neighborhood Mea Shearim, associated with the ultra-Orthodox society, and the adjacent neighborhoods to the north and west, are expected to suffer increased ground shaking.
An instrumentally located epicenter (1992, M>3) is marked by a solid black square (www.gii.co.il).
The Medvedev–Sponheuer–Karnik (MSK) intensity scale evaluates the severity of ground shaking for a given location; the mean seismic intensity in Jerusalem during the 1927 6.2 M Jericho earthquake ranged 6.5-7.5 in MSK scale (Avni, 1999; Shapira et al., 1993; Hough and Avni, 2011). This scale corresponds roughly to the MMS given at the bottom of Table 1.

**Appendix A – Quantitative Questionnaire**


1.   Gender (male/female)
2.   Age ______
3.   City/town of residence ___________
4.   Neighborhood of residence __________________
5.   Marital status

1.   Single

2.   Married

3.   Divorced

4.   Widowed

6.   Do you have children? Yes/No

If Yes, number of children _______ at ages _________

7. What, in your opinion, is the likelihood that a disastrous earthquake will occur in Israel within the next five

803    years?

1.   No chance

2.   Low likelihood

3.   Medium likelihood

4.   High likelihood

5.   Certainly

8.  What, in your opinion, is the likelihood that a disastrous earthquake will occur in your area within the next

five years?

1.   No chance

2.   Low likelihood

3.   Medium likelihood

4.   High likelihood

5.   Certainly

9. To what extent do you agree with the following statements?

|   |   | Totally disagree | Agree to an extent | Quite agree | Much agree | Very much agree | Do not know | Irrelevant |
|---|---|---|---|---|---|---|---|---|
| 1 | I do not believe that an earthquake will occur in my area and therefore do not need to prepare | | | | | | | |

| | | | | | | | |
|---|---|---|---|---|---|---|---|
| 2 | Regarding an earthquake, I think "what will be will be," hence I do not prepare | | | 35 | | | | |
| 3 | I think that we need to do everything possible in order to prepare for an earthquake | | | | | | | |
| 4 | The forecasted scenarios following an earthquake are exaggerated, therefore it is unnecessary to prepare | | | | | | | |
| 5 | I think that, even if there will be an earthquake in my place, improvisations may solve the situation | | | | | | | |
| 6 | Even if an earthquake occurs, I will know what to do - we have been adequately exposed to emergencies in this country | | | | | | | |
| 7 | The idea of an earthquake is so frightening - I | | | | | | | |

| | | | | | | | |
|---|---|---|---|---|---|---|---|
| | avoid thinking of it | | | | | | |
| 8 | The chance to survive an earthquake is so slight, that preparing is not worthwhile | | | | | | |
| 9 | I feel that I will be able to deal with an earthquake situation | | | | | | |
| 10 | Earthquake damages are avoidable with "proper" preparation | | | | | | |
| 11 | I have the knowledge and aptitude that will help me mitigate in case of an earthquake | | | | | | |
| 12 | I do what it takes to prepare for the possibility of an earthquake even if it costs money and requires time | | | | | | |
| 13 | I actively look for information regarding earthquake preparedness | | | | | | |

| | | | | | | | | |
|---|---|---|---|---|---|---|---|---|
| 14 | Earthquake preparedness is a religious obligation enforced by the directive that one must guard one's own life (hishtadlus) | | | | | | | |
| 15 | Earthquake preparedness suits people with deficient beliefs | | | | | | | |
| 16 | Should there be organized instruction in the community, I would like to learn how to prepare for an earthquake | | | | | | | |


10. Does your apartment provide a sheltered space?
6. Yes
7. No
8. Irrelevant
11. Houses erected in Israel before 1995 were not constructed according to the contemporary Earthquake Code.
Does your home pass that Earthquake Code?
9. Yes
10. Probably yes
11. Probably not
12. No
13. Do not know
12. Does your home pass the Earthquake Code of 1980?
14. Yes
15. Probably yes
16. Probably not
17. No
18. Do not know
13. Were the foundations of your home retrofitted, e.g. under State Master Plan 38?

| 836 | 19. Yes |
| 837 | 20. No |
| 838 | 21. Do not know |
| 839 | 22. Irrelevant |
| 840 | 14. Have you or others insured your home against earthquake hazard? |
| 841 | 23. Yes |
| 842 | 24. No |
| 843 | 25. Do not know |
| 844 | 26. Irrelevant - I do not own a home |
| 845 | 15. To the best of your knowledge, what measures can a family take to prepare for an earthquake? |
| 846 | _______________________________________ |
| 847 | 16. In preparation for an earthquake one is required to stabilize bookshelves and fasten them to walls; support |
| 848 | water tanks, gas tanks and air-conditioners; store toxics and inflammables away from heat sources; keep heavy |
| 849 | objects close to the floor. Have you prepared according to all, or some, of the above precautions? |
| 850 | 27. Yes |
| 851 | 28. No |
| 852 | 29. Partially. Please detail: _______________________________________ |
| 853 | 17. Do you have emergency equipment available in your home, such as a first aid kit, water and canned food, |
| 854 | medicines, battery-powered flashlight and radio, etc.? |
| 855 | 1. Yes |
| 856 | 2. No |
| 857 | 3. Some of the equipment. Please specify: _________________ |
| 858 | 18. What will you do in case (Heaven forfend) that you feel an earthquake? |
| 859 | ___________________________________________________________ |
| 860 | 19. Do you and your family know the safety guidelines for behavior in the event of an earthquake (such as |
| 861 | switching off electricity and gas lines, exiting to an open area away from buildings or entering a safe room or |
| 862 | stairwell)? |
| 863 | 1. Yes |
| 864 | 2. No |
| 865 | 3. Partially. Please specify _______________________________________ |
| 866 | 20. Are you and your household members acquainted with the safety instructions for an earthquake? |
| 867 | 30. Yes |
| 868 | 31. No |
| 869 | 32. Partially. Please detail: _______________________________________ |
| 870 | 21. (for parents of minors) |
| 871 | Have you discussed the safety instructions for an earthquake with your children? |
| 872 | 33. Yes |
| 873 | 34. No |
| 874 | 35. Partially. Please detail: _______________________________________ |
| 875 | 22. Have you practiced with your children proper behavior during an earthquake? |

36. Yes
37. No
38. Partially. Please detail: _________________________________________
23. Overall, have you prepared for the possibility of an earthquake in Israel?
39. Not at all
40. To a minor extent
41. To a moderate extent
42. To a large extent
43. To a very large extent
44. Do not know
24. Occasionally earthquake protection instructions are published in the various media. Have you come across
such publications?
1. Yes
2. No
25. If so, where did you come across a publication on the subject?
1. Newspapers
2. Direct mail
3. Radio
4. Websites
5. Other _________________
26. What do you remember from the publication you saw or heard? _______________________________________
27. In your view, is the fact that instructions arrive from the military authorities significant for the ultra-
Orthodox public? _____________________________________________
28. To the best of your knowledge, is there a religious duty to prepare for an earthquake?
_________________________________________
29. In your opinion, how can awareness for earthquake preparedness be raised in the ultra-Orthodox public?
_________________________________________
30. Will a Halakhic ruling or a rabbi instruction convince you to prepare for an earthquake?
_________________________________________
31. What are the strong aspects of your community which might assist the entire community to function under
an emergency? _________________________________________
32. The solidarity in the ultra-Orthodox society is commendable. How in your view is it possible to translate this
solidarity for promoting the earthquake preparedness of the ultra-Orthodox public?
______________________________________________________
33. Education
Male: (secondary/tertiary/academic/Yeshiva)
Female: (primary/secondary/tertiary/academic)
34. Community you belong to _________________________
35. Your estimate of your economic status:
45. Below average
46. Average
47. Above average
48. Refuse to answer
36. What type of mobile phone do you typically possess:
49. Kosher
50. Basic with texting
51. Supports Kosher
52. Protected smartphone
53. Unprotected smartphone
54. I do not hold a cellular phone
37. Is there anything else you would like to add regarding earthquake preparedness in Israel?
________________________________________________

**Appendix B – Qualitative Interview Guide**


Note: This basic interview guide (qualitative questionnaire) includes issues and questions that are common to a
broad range of interviewees. We adapted each interview guide according to the specific interviewee's
organization, professional field, expertise, etc.

1. Can you please give some background about your organization?
2. In which activities does your organization participate regarding earthquake preparedness?
3. Does your organization promote preparedness for earthquakes in the ultra-Orthodox society as well? If so, I
would appreciate hearing about such activities. What do they include?
4. Is the activity in the ultra-Orthodox society similar to the activities in other societies? Does the work with
this society require adjustments?
5. Are you aware of special challenges in the work with the ultra-Orthodox public? If so, what are these
challenges?
6. Are you aware of special opportunities in the work with the ultra-Orthodox public? If so, what are these
opportunities?
7. What, in your opinion, can be improved or strengthened in the work of your organization with the ultra-
Orthodox public?
8. How, in your opinion, can the awareness for earthquake preparedness be elevated within the ultra-Orthodox
public?
9. Is it significant in the ultra-Orthodox public that the instructions on this topic arrive from the Home Front
Command and the state authorities?
10. To the best of your knowledge, what is the Halakha's (the collective body of Jewish religious laws) position
regarding earthquake preparedness?
11. In your experience, how does the Jewish Halakha and the faith of ultra-Orthodox Jews affect their actual
preparedness?
12. In your view, which characteristics of the ultra-Orthodox society assist or impede its functioning under a
state of emergency?
13. The ultra-Orthodox society is characterized by strong solidarity. Do you view it feasible to harness this
solidarity to promote the preparedness of the ultra-Orthodox public for an earthquake? If so, how?
14. Is there anything you wish to add regarding the topic of this research? Anything that I did not ask about and
is important for us to know?
15. Whom else would you recommend us to talk with?
16. We would like to talk with relevant key figures in the ultra-Orthodox community. Are there people in the
ultra-Orthodox society whom you work with and who might assist us?

# Appendix C – Quantitative Findings

Level of belief regarding the occurrence of an earthquake

|  | No chance (1) | Low likelihood (2) | Medium likelihood (3) | High likelihood (4) | Certainly (5) | Mean | SD |
|---|---|---|---|---|---|---|---|
| What, in your opinion, is the likelihood that a disastrous earthquake will occur in Israel within the next five years? | 17% | 38% | 33% | 11% | 1% | 2.39 | 0.927 |
| What, in your opinion, is the likelihood that a disastrous earthquake will occur in your area within the next five years? | 22% | 42% | 24% | 12% | 0% | 2.24 | 0.928 |

To what extent do you agree with the following statements?

|  | Totally disagree | Agree to an extent | Quite agree | Much agree | Very much agree | Do not know |
|---|---|---|---|---|---|---|
| I do not believe that an earthquake will occur in my area and therefore do not need to prepare | 59% | 17% | 12% | 8% | 3% | 1% |
| Regarding an earthquake, I think "what will be will be," hence I do not prepare | 67% | 15% | 9% | 5% | 3% | 1% |
| The forecasted scenarios following an earthquake are exaggerated, therefore it is unnecessary to prepare | 63% | 16% | 9% | 7% | 4% | 1% |
| The chance to survive an earthquake is so slight, that preparing is not worthwhile | 80% | 14% | 2% | 1% | 2% | 1% |
| Earthquake preparedness is a religious obligation enforced by the directive that one must guard one's own life (hishtadlus) | 17% | 11% | 26% | 22% | 24% | 0% |
| I think that, even if there will be an earthquake in my place, improvisations may solve the situation | 52% | 20% | 12% | 8% | 5% | 3% |
| Even if an earthquake occurs, I will know what to do - we have been adequately exposed to emergencies in this country | 48% | 24% | 13% | 7% | 4% | 4% |
| I feel that I will be able to deal with an earthquake situation | 24% | 20% | 23% | 21% | 6% | 6% |
| I have the knowledge and aptitude that will help me mitigate in case of an earthquake | 31% | 28% | 23% | 12% | 4% | 2% |

| | | | | | |
|---|---|---|---|---|---|
| I do what it takes to prepare for the possibility of an earthquake even if it costs money and requires time | 53% | 22% | 12% | 7% | 4% | 2% |
| I actively look for information regarding earthquake preparedness | 66% | 15% | 10% | 4% | 1% | 4% |


| | Yes | No |
|---|---|---|
| In preparation for an earthquake one is required to stabilize bookshelves and fasten them to walls; support water tanks, gas tanks and air-conditioners; store toxics and inflammables away from heat sources; keep heavy objects close to the floor. Have you prepared according to all, or some, of the above precautions? | 23% | 77% |
| Do you have emergency equipment available in your home, such as a first aid kit, water and canned food, medicines, battery-powered flashlight and radio, etc.? | 48% | 52% |
| Do you and your family know the safety guidelines for behavior in the event of an earthquake (such as a switching off electricity and gas lines, exiting to an open area away from buildings or entering a safe room or stairwell)? | 51% | 49% |
| Have you discussed the safety instructions for an earthquake with your children? | 15% | 85% |
| Have you practiced with your children proper behavior during an earthquake? | 3% | 97% |


| | Yes / Probably yes | Probably not / No | Do not know |
|---|---|---|---|
| Houses erected in Israel before 1995 were not constructed according to the contemporary Earthquake Code. Does your home pass that Earthquake Code? | 39% | 49% | 12% |
| Does your home pass the Earthquake Code of 1980? | 47% | 36% | 17% |


| | Not at all | To a minor extent | To a moderate extent | To a large / a very large extent |
|---|---|---|---|---|
| Overall, have you prepared for the possibility of an earthquake in Israel? | 45% | 31% | 18% | 6% |


| | Yes | No |
|---|---|---|
| Occasionally earthquake protection instructions are published in the various media. Have you come across such publications? | 46% | 54% |


| | Newspapers | Websites | Direct mail | Radio |
|---|---|---|---|---|
| If so, where did you come across a publication on the subject? | 41% | 31% | 8% | 20% |

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
