# Peer review of "The case of the Jewish ultra-Orthodox society in Israel"

_Natural Hazards and Earth System Sciences, 2019_

## Referee Comment (RC1) · Anonymous Referee #1 · 11 Feb 2020

GENERAL COMMENT

The paper examines the effects of religious beliefs and customs on earthquake preparedness among the Jewish ultra-orthodox community in Israel. This is a very interestig and original subject within the topic of the difficult seismic preparedness of the general public.

However, I think that the paper needs to be reworked; for these reasons, I suggest a new submission after major revisions and hope that these few lines below can help.

SPECIFIC COMMENTS

Abstract : the abstract outlines the findings of the survey and interviews, but the last sentence does not outline any new idea.

1. Introduction Please rewrite with more specific and more precise data and observations.

2.1 Earthquake preparedness: due to the title of the paper, this part should be rewritten to focus only on religious communities of the world or of the region, and not on 'marginalized minority groups'.

2.2 The Jewish Ultra-Orthodox Sector: does the word "sector" mean that the JUO community leave in a specific area? As it is not well known in Europe for instance, please also precise if the JUO have a specific language, etc.

2.3 Earthquake Hazard in Jerusalem Please add a figure showing Israel, Jerusalem, and fault lines with dots or stars illustrating historic and instrumental earthquakes with intensities or magnitudes. Make a zoom on the eventual JUO sector of Jerusalem and eventual seismic zoning (if it does exist) and eventual shaking maps (if they do exist): the aim of this demand is to illustrate the specific vulnerability of the people and buildings.

3.1 Survey How many questionnaires were sent OR give a percentage of responses (can help distinguishing persons interested in preparedness who could later be used as vectors of information), see line 337

4 Results Please simplify and re-order the answers. Help the reader with graphs, pie charts etc. Outline the answers of the religious leaders and key figures of the JUO community, for they could become vectors of preparedness information.

Family names of safety officers are cited, is it common and accepted by the so-called persons, and is it allowed by the NHESS journal?

5 Discussion and Conclusions The discussion needs to be reworked, also focusing on religious minorities. The SWOT analyse is a very good idea that could be more

developed to offer preparedness perspectives based on the strong social capital of the JUO. A good perspective would be to submit preparedness ideas to interviewed religious leaders: then emphasis could be put on advices in hardware stores, simple and cheap home security (furniture organisation and securing, basic rescue kits for the family), practical skills based on the specific neighbourhood solidarity in the JUO, the organisation of participatory experiences by the religious authorities for instance.

Figure 1 : in an inset, show map of Israel + Jerusalem location + any specific JUO sector Table 1 : precise 'epicentral' or 'maximal' or 'downtown' or 'whatever' intensity

---

## Referee Comment (RC2) · Anonymous Referee #2 · 24 Apr 2020

Erblich et al. study ". . . the effects of religious beliefs and customs on earthquake preparedness among the Jewish ultra-orthodox community in Israel, a significant religious minority with unique social, cultural, and economic characteristics." They conducted a survey and in-depth interviews among 228 members of this group and concluded that despite the long-documented history of damaging earthquakes in the Levant area, the majority of that community have a low level of hazard knowledge, awareness and preparedness. Based on their findings, Erblich et al. discuss the reasons and factors that hinder preparedness among the examined group and suggest ideas and ways to cope with these difficulties in order to improve awareness and upgrade readiness among these people.

[Figure]

This is a very important study that points to a severe lacuna in earthquake awareness and preparedness within the ultra-orthodox community of Israel. Although the study focuses on a specific group, its outcomes and understandings can be of interest to elsewhere places and other societies. However, the manuscript is not yet ready for publication. It needs a major revision and another cycle of full review. Hereby I will explain my opinion, what is needed to improve the manuscript and prepare it for publication.

Earthquake hazards and damage in Jerusalem

Section 2.3 does not present the state of the art knowledge and understandings regarding earthquake hazards and damage in Jerusalem. For example, Avni (1999) presented detailed lists and reports of what had happened in Jerusalem during the 1927 earthquake; Salamon et al. (2010) conducted earthquake hazard evaluation for Jerusalem; Zohar et al. (2017) presented updated list of earthquakes and the damage they cause in Israel, including in Jerusalem. With this respect, Section 2.3 and Table 1 should be revised thoroughly.

Data

Overall, the raw data is not presented and the little that is shown does not allow the reader to follow and repeat the results and the final conclusions. The raw data should be presented in full, in tables and diagrams. Quantitative analysis: 1. There is a need to present the online questionnaire in full, may be as an appendix.

2. Please present the demography of those who responded and filled the questionnaire, i.e. the information relevant to this study such as age, gender, income, education, area of living, social subgroup, etc. This of course should be kept within the confidentiality and anonymity promised to the reviewees, but still allow sufficient transparency of the database. Section 4.1.1 is not enough.

3. Please report how many forms were distributed against how many were replied and

filled.

4. The findings from the questionnaires should also be presented in full as a background for the analysis of the results and the discussion.

Qualitative analysis 1. Even though the interviews were open, there is a need to show the main leading issues and questions, as well as the answers, at least in a schematic way. Otherwise, there is no way to infer and conclude the systematic attitude or approach of the examined population.

2. The demography of those who participated in the open interview should also be presented. Which segment of the ultra-orthodox group do they belong to? Is this segment represent the entire ultra-orthodox group?

3. Please note how many people were approached against how many were interviewed.

Lines 256-7: "We then created a shortened version of the questionnaire to spread via an online form." However, in lines 371-2 it is said: ""Most people do not own a television at home, and many do not have access to the internet or the radio". The reader may understand that those who replied online does not represent that group?

Analysis of data and Results

It is not possible to evaluate how much the level of knowledge and awareness of the ultra-orthodox group differs from the 'reference standard' of Israel. There is a need to present the 'common' level of knowledge in Israel, and by comparison, show the difference (sections 4.1.2, 4.1.3). Likewise, there is a need to show that the group of interviewees (228 responses) does represent the ultra-orthodox group (one million), so that the results of this study can be generalized and represent the group. Some sections or paragraphs seem to belong to the Introduction rather than to the results, e.g. lines 382-390, 402-404. In any case, such facts or statements (e.g. lines 389–90: "... most ultra-orthodox schools still do not allow military personnel to conduct

trainings") should be supported by a proper source. Sometimes it is not clear whether the given statement refers to a direct outcome of this study? For example, lines 395-398 regarding the "Low Socioeconomic Status". Lines 442 – 444: belong to technology section in lines 481?

Section 5 Discussion and Conclusions

Section 5 is hard to follow. There are issues that belong to the introduction, discussion, conclusions and recommendations, and I got confused which is what. For example, the content in lines 491-495 sounds reasonable, but is it a result of this work or just ideas taken from the given references. Sometimes the phrasing seems to 'hide' the finding of this study, for example, lines 489-491: "Past research has much to say regarding how emergency preparedness among marginalized communities can be improved, including both conduct during emergencies as well as recovery efforts following disasters. Many of our findings support this literature." I think that this should be rephrased the other way around: first state in your own words what you have found and then discuss whether it supports (or not…) past research and give the appropriate reference. The main understandings, outcomes and suggestions are scattered along the text and thus I would recommend splitting Section 5 into three: Discussion, Conclusions and Recommendations.

Conclusions

I recommend summarizing the actual findings of this work as independent statements, rather than mentioning that: "Many of our findings support this literature…".

Recommendations

Although the recommendations (e.g. lines 549-55) are part of the conclusions, I suggest listing it also in bullets or in a table, so as the authorities, who would (or should) be interested to implement the outcomes of this work, get to the point simply and easily.

Oversimplification of the situation?

Lines 514-15 state that "Given that the ultra-orthodox constitute a religious minority in Israel, the state authorities' representation of these communities is minimal" The status of the ultra-orthodox minority in Israel is unique. On one hand, they are a significant part of the government, sometimes even the balancing power, and are able to achieve much influence, beyond their relative share in the population. On the other hand, they refrain from accepting the Israeli authorities as is, and adhere to their own community leaders. They seem to prefer to isolate themselves and maintain their own way of life. In that sense, once you conclude and recommend that (line 554-5): "In addition, community leaders should be asked to give their support before approaching the community they lead", it means (in the subtext) that the community leaders may share the responsible for the "... low level of hazard knowledge and a high level of disbelief that a devastating earthquake would occur in their area in the near future." (lines 39-40). As a result, a possible recommendation could be that instead of authorizing "... local non-governmental organizations official responsibilities in the area of earthquake... " (line 553), the local community leaders should authorize the formal Israeli authorities the responsibilities in the area of earthquake. This is a very complex situation and it means that improving the level of knowledge and awareness in this minority group is not just a simple list of 'to do'.

Referencing

Many statements sounds reasonable, however, they need to be supported by a reference or source of data, e.g. line 90; lines 209-210: "With regard to earthquake preparedness, most members of the Jewish ultra-orthodox society in Israel live in ... buildings that do not meet the standards for earthquakes."

References not mentioned in the text

Avni, R., 1999. The 1927 Jericho Earthquake, Comprehensive Macroseismic Analysis Based on Contemporary Sources. (PhD.). Ben Gurion University, Beer-Sheva. Salamon, A., Katz, O. and Crouvi, O. (2010). Zones of required investigation for

earthquake-related hazards in Jerusalem. Natural Hazards, 53: 375-406. Zohar, M., Salamon, A. and Rubin, R., (2017). Earthquake damage history in Israel and its close surrounding - evaluation of spatial and temporal patterns. Tectonophysics, 696: 1-13.
* * *

---

## Author Comment (AC1) · 13 Jun 2020

June 12, 2020

**Authors' response to Referee 1**

We would like to thank Referee 1 for the exceptionally constructive feedback. We found the comments and suggestions offered by the referee very helpful and we appreciate the opportunity to revise key aspects of the article. We have made all of the changes suggested by the referee, and we believe that the revisions create a stronger and clearer paper.

**Referee comment: Abstract**

The abstract outlines the findings of the survey and interviews, but the last sentence does not outline any new idea.

**Authors' response:**

We concluded the abstract by stating the uniqueness of this research: it studies a religious group that is also a cultural minority compared to the secular authorities and therefore requires special adaptations. It offers a perspective on the complex reality of hazard preparedness in a religiously diverse country. We also listed some of the key adaptations suggested to policy makers.

**Referee comment: Introduction**

Please rewrite with more specific and more precise data and observations.

**Authors' response:**

We rewrote the introduction using the following structure: Information on earthquake destruction, citing Bozorgnia and Bertero, 2004; The importance of considering the needs of religious groups in earthquake management, citing Baytiyeh and Naja, 2014; Sun et al., 2018, Gianisa and Le De, 2018; The goal of our research; The research questions; Summary of the main arguments.

**Referee comment: Earthquake preparedness among marginalized minority groups and religious communities**

Due to the title of the paper, this part should be rewritten to focus only on religious communities of the world or of the region, and not on 'marginalized minority groups'.

**Authors' response:**

We rewrote this section focusing mainly on the advantages and disadvantages of religious groups in disaster management, as described in past literature. We discussed the impact of religion on the three phases of an earthquake: the preparation phase, the emergency phase and the restoration phase. We presented suggestions from past literature on how to improve the three stages of an earthquake in religious communities and in the general populace. We added data citing additional sources such as Drabek, 2001; Sattler et al., 2000; Wilkin et. al, 2019, and others.

Since the social group that we study is also a minority in its country, we listed some characteristics of minorities which have a great impact on disaster preparedness.

**Referee comment: The Jewish Ultra-Orthodox Sector**

Does the word "sector" mean that the JUO community live in a specific area? As it is not well known in Europe for instance, please also precise if the JUO have a specific language, etc.

**Authors' response:**

Most ultra-orthodox people live either in towns of their own or in closed community neighborhoods within diverse cities.

Although some of the communities still speak Yiddish as their first language, most of the ultra-orthodox people are Hebrew speakers like most Israelis.

We added this information to the manuscript.

**Referee comment: Earthquake Hazard in Jerusalem**

Please add a figure showing Israel, Jerusalem, and fault lines with dots or stars illustrating historic and instrumental earthquakes with intensities or magnitudes. Make a zoom on the eventual JUO sector of Jerusalem and eventual seismic zoning (if it does exist) and eventual shaking maps (if they do exist): the aim of this demand is to illustrate the specific vulnerability of the people and buildings

**Referee comment: Figure 1**

in an inset, show map of Israel + Jerusalem location + any specific JUO sector Table 1: precise 'epicentral' or 'maximal' or 'downtown' or 'whatever' intensity

**Authors' response:**

We added two figures which demonstrate the specific vulnerability of the people and buildings of the ultra-orthodox sector:

Figure 1 - instrumental earthquakes (>3M) were added (Fig. 1b); historical earthquakes are presented in Fig. 1a. Moreover, in Fig. 1b the dominant faults are marked, and settlements with ultra-orthodox residents are presented.

Figure 2 - map of Jerusalem with ultra-orthodox neighborhoods. Calculated seismic intensities (Avni, 1999) and zone of ground amplification (Salamon et al., 2010) are presented. Instrumental earthquakes >2M in Jerusalem are also presented.

Please view the figures at the end of this letter.

**Referee comment: Survey**

How many questionnaires were sent OR give a percentage of responses (can help distinguishing persons interested in preparedness who could later be used as vectors of information), see line 337

**Authors' response:**

The response rate was around 90% for the in-person interviews. We do not know the response rate to the online questionnaire since we did not control its distribution.

**Referee comment: Results**

Please simplify and re-order the answers.

**Referee comment: Quantitative findings**

Help the reader with graphs, pie charts etc.

**Authors' response:**

We included diagrams to illustrate the main findings, e.g., we added a diagram to summarize the level of belief of the ultra-orthodox people regarding the occurrence of a disastrous earthquake in the near future (please see Figure 3 at the end of this letter).

We elaborated on the demographics of the respondents (age, gender, marital status, income, education, area of living, social subgroup and type of phone they own). We also added data regarding the level of preparedness of the community. Furthermore, we added the translated questionnaire and a table of the full findings as an appendix. Consequently, we believe that it is now easier to follow the results.

**Referee comment: Qualitative findings**

- Outline the answers of the religious leaders and key figures of the JUO community, for they could become vectors of preparedness information.

- Family names of safety officers are cited, is it common and accepted by the so-called persons, and is it allowed by the NHESS journal?

**Authors' response:**

- The answers of the respondents are summarized in a revised SWOT analysis and in clearer and revised points in the conclusion section. Our recommendations are deducted from the answers of the religious leaders and key figures that we interviewed.

- All the research participants' names in the manuscript are pseudonyms and appear there for simplification purposes only. We clarified this in the text.

We enclosed the interview guide, which was written to help the interviewers frame the interviews and keep them on track, as an appendix.

**Referee comment: Discussion and Conclusions**

The discussion needs to be reworked, also focusing on religious minorities. The SWOT analyze is a very good idea that could be more developed to offer preparedness perspectives based on the strong social capital of the JUO. A good perspective would be to submit preparedness ideas to interviewed religious leaders: then emphasis could be put on advices in hardware stores, simple and cheap home security (furniture organization and securing, basic rescue kits for the family), practical skills based on the specific neighborhood solidarity in the JUO, the organization of participatory experiences by the religious authorities for instance.

**Authors' response:**

We split the section "Discussion and Conclusion" into three: "Discussion", "Conclusion" and "Recommendations".

In the discussion we focused on religious minorities as suggested. We reviewed the level of awareness and preparedness of the ultra-orthodox sector and compared the findings with the literature. We discussed the community's unique characteristics and stated how they may increase preparedness or hamper it.

In the discussion we brought up an interesting point regarding retrofitting: Half of our survey respondents indicated that they do not live in buildings that meet the legal safety standards despite the fact that some of their neighborhoods are in areas with increased ground shaking during an earthquake (Salamon et al., 2010) (Fig. 2). Alarmingly, none of our respondents recommended retrofitting, a crucial strategy for minimizing the harm caused by earthquakes (Bozorgnia and Bertero, 2004). Our interviews with relevant stakeholders further confirmed that very few ultra-orthodox people are interested in retrofitting. They explain that the reasons for this include a lack of awareness regarding three factors: the potential impact of earthquakes, the significance of building conditions in reducing damage, and the government's willingness to support retrofitting. Additional reasons are the intangibility of the danger and the low economic status of ultra-orthodox society. These findings stand in agreement with Maldonado et al. (2016) and Lucini (2014), who found that minority groups show a low level of self-protection and preparedness, a low level of hazard knowledge and a low level of action during the emergency phase.

In the discussion, we compared the level of knowledge and readiness to prepare for an earthquake of the ultra-orthodox population to that of the general Israeli public, using the findings of Ya'ar et al. (2015) regarding the Israeli population. Please see Figure 4 at the end of this letter.

The research findings are summarized in the "conclusion" section. Our research questions included three main themes: the actual state of earthquake preparedness in the ultra-orthodox sector, characteristics that may hinder or promote preparedness, and ways of improving preparedness. The findings from the first two questions are summarized in a SWOT analysis that we further developed in accordance with the referee's recommendation, emphasizing the potential contribution of the ultra-orthodox society's strong social capital. The findings from the third question are summarized in bullets within the section.

We listed our recommendations in a separate section using a clear bullet format. We included the referee's important and creative ideas in the list.

To conclude, we addressed all the issues raised by Referee 1. We would like to thank the referee again for the work invested in reviewing our manuscript. We truly believe that the review process substantially contributed to the article and we hope that the current version will be accepted to *Natural Hazards and Earth System Sciences*.

Sincerely,

The authors

**Figures**

[Figure]

Figure 1:

(a)- Estimated spatial extent of ruptures from historic periods along the DSFS (after Agnon, 2014). The position of the events in the graph is projected from the map of the DSFS below in (b).

(b) - The DSFS main branches (Hamiel et al., 2018; Hofstteter et al., 1996; Kagan et al., 2011; Politi, 2011; Sharon et al., 2018) over instrumental earthquakes record (from 1984) of >3 Md (www.gii.co.il).

Settlements with Ultra-Orthodox residents are marked; the majority is in Jerusalem and Bnei Brak. [1] – Beit Shemesh, [2] – Beitar Illit, [3] – Modi'in Illit.

[Figure]

**Figure 2:**

**Map of neighborhoods in Jerusalem with Ultra-Orthodox residents (modified after Community administrators Jerusalem map, 2009; Golan, 2020) over ground amplification map (Salamon et al., 2010).**

[1] – Har Nof, [2] – Bait Vegan, [3] _ Kiryat Hayovel, [4] – Ramot, [5] – Romema, [6] - Givat Mordechai, [7] - French Hill, [8] - Mekor Baruch, [9] - Ramat Eshkol, [10] - Sha'arei Hesed, [11] - Zikhron Moshe, [12] – Katamon and Katamonim, [13] – Arnona, [14] – Bukharim, [15] – Morasha, [16] - Shmuel HaNavi.

**Mea Shearim, and the neighborhoods adjacent to it to the north are associated with the ultra-orthodox sector. They are in an area marked by increased ground shaking during an earthquake.**

**Instrumental earthquakes (from 1984) of <2 Md are presented by black rectangles together with the year of quaking (www.gii.co.il). Medvedev–Sponheuer–Karnik scale (MSK) intensity scale evaluates the severity of ground shaking in an area of earthquake occurrence; the mean seismic intensity in Jerusalem shows strong-to very strong shaking zones from the 1927 6.2 M Jericho earthquake (Avni, 1999).**

Level of belief that a disastrous earthquake will occur in Israel/in your area in the next five years

[Figure]

**Figure 3:**

Level of belief that a disastrous earthquake will occur in Israel/in your area in the next five years

[Figure]

**Figure 4:**
Level of knowledge and readiness to prepare for an earthquake in Israel

---

## Author Comment (AC2) · 13 Jun 2020

June 12, 2020

**Authors' response to Referee 2**

We would like to thank Referee 2 for the exceptionally constructive feedback. We found the comments and suggestions offered by the referee very helpful and we appreciate the opportunity to revise key aspects of the article. We have made all of the changes suggested by the referee, and we believe that the revisions create a stronger and clearer paper.

**Referee comment: Earthquake Hazard in Jerusalem**

Section 2.3 does not present the state-of-the-art knowledge and understandings regarding earthquake hazards and damage in Jerusalem. For example, Avni (1999) presented detailed lists and reports of what had happened in Jerusalem during the 1927 earthquake; Salamon et al. (2010) conducted earthquake hazard evaluation for Jerusalem; Zohar et al. (2017) presented updated list of earthquakes and the damage they cause in Israel, including in Jerusalem. With this respect, Section 2.3 and Table 1 should be revised thoroughly.

**Authors' response:**

The detailed information of Avni, 1999 (written in Hebrew) was translated and summarized in Hough & Avni (2011) which we cited. Following the referees' suggestions, we use some of the detailed information from Avni (1999) together with the work of Salamon et al. (2010). The figures were updated using the recommended articles:

Figure 1 - instrumental earthquakes (>3M) were added (Fig. 1b); historical earthquakes are presented in Fig. 1a. Moreover, in Fig. 1b the dominant faults are marked, and settlements with ultra-orthodox residents are presented.

Figure 2 - map of Jerusalem with ultra-orthodox neighborhoods. Calculated seismic intensities (Avni, 1999) and zone of ground amplification (Salamon et al., 2010) are presented. Instrumental earthquakes >2M in Jerusalem are also presented.

Please view the figures at the end of this letter.

**Referee comment: Survey**

- Lines 256-7: "We then created a shortened version of the questionnaire to spread via an online form." However, in lines 371-2 it is said: "Most people do not own a television at home, and many do not have access to the internet or the radio". The reader may understand that those who replied online do not represent that group?

- Likewise, there is a need to show that the group of interviewees (228 responses) does represent the ultra-orthodox group (one million), so that the results of this study can be generalized and represent the group.

**Authors' response:**

- Regarding internet access, 48% of the respondents had no smartphone for religious reasons. However, 32% of non-smartphone owners answered the online form. This implies that many ultra-orthodox people have, albeit limited, internet access and are also represented in the online form. This information was added to the quantitative findings.

It should be noted that the limited access to the internet was one of the considerations for choosing in-person interviews for the majority of the respondents.

- In this research we used non-probability sampling that may not represent the entire ultra-orthodox population in Israel. There were only 288 respondents, there was an over-representation of men, the average age was young, and the great majority of respondents lived in Jerusalem and its surrounding areas. However, in our opinion, there are also considerable advantages to the sample that we chose. Firstly, the in-person interview surveys were conducted by ultra-orthodox or orthodox surveyors, enabling us to reach out to ultra-orthodox people who would probably refuse to respond to a telephone survey from an academic institution. A telephone survey which is faster and simpler might involve a considerable bias when it comes to the ultra-orthodox public. Secondly, the survey conducted in face-to-face interviews allowed a very long and detailed questionnaire to be answered, it allowed elaboration on questions and answers, it ensured that questions and respondents were understood, and it avoided offhand answers. None of this happens in a telephone or internet survey. Thirdly, focusing on particular communities, like ultra-orthodox communities in Jerusalem and its surrounding areas, allows a deep understanding of their needs and challenges and enables possibility of working with them over time. For example, it allows efficient training in these communities, like the training that dozens of our ultra-orthodox students currently lead in several ultra-orthodox communities in Jerusalem, after we had trained these students. Lastly, the young people that were interviewed are, or will soon become parents of children. We have seen from the literature that children's education is one of the most impactful tools in earthquake preparedness. From this point of view, we also wanted to co-opt them.

**Referee comment: Results (general comment)**

Overall, the raw data is not presented and the little that is shown does not allow the reader to follow and repeat the results and the final conclusions. The raw data should be presented in full, in tables and diagrams.

**Author response:**

Please see the two subsequent responses below.

**Referee comment: Quantitative findings**

Quantitative analysis:

1. There is a need to present the online questionnaire in full, maybe as an appendix.

2. Please present the demography of those who responded and filled the questionnaire, i.e. the information relevant to this study such as age, gender, income, education, area of living, social subgroup, etc. This of course should be kept within the confidentiality and anonymity promised to the reviewees, but still allow sufficient transparency of the database. Section 4.1.1 is not enough.

3. Please report how many forms were distributed against how many were replied and filled.

4. The findings from the questionnaires should also be presented in full as a background for the analysis of the results and the discussion.

5. It is not possible to evaluate how much the level of knowledge and awareness of the ultra-orthodox group differs from the 'reference standard' of Israel. There is a need to present the 'common' level of knowledge in Israel, and by comparison, show the difference (sections 4.1.2, 4.1.3).

**Authors' response:**

We thank the referee for this helpful comment; we have added all that was recommended.

We included diagrams to illustrate the main findings, e.g., we added a diagram to summarize the level of belief of the ultra-orthodox people regarding the occurrence of a disastrous earthquake in the near future (please see Figure 3 at the end of this letter).

We have updated the demographics of the respondents (age, gender, marital status, income, education, area of living, social subgroup and type of phone they own). We also added data regarding the level of preparedness of the community. Furthermore, following the referee's advice, we added the questionnaire and a table of the full findings as an appendix.

The response rate was around 90% for the in-person interviews. We do not know the response rate to the online questionnaire since we did not control its distribution. We added this information to the manuscript.

In the discussion, we compared the level of knowledge and readiness to prepare for an earthquake of the ultra-orthodox population to that of the general Israeli public, using the findings of Ya'ar et al. (2015) regarding the Israeli population (please see Figure 4 at the end of this letter).

**Referee comment: Qualitative findings**

1. Even though the interviews were open, there is a need to show the main leading issues and questions, as well as the answers, at least in a schematic way. Otherwise, there is no way to infer and conclude the systematic attitude or approach of the examined population.

2. The demography of those who participated in the open interview should also be presented. Which segment of the ultra-orthodox group do they belong to? Does this segment represent the entire ultra-orthodox group?

3. Please note how many people were approached against how many were interviewed.

Some sections or paragraphs seem to belong to the Introduction rather than to the results, e.g. lines 382-390, 402-404. In any case, such facts or statements (e.g. lines 389– 90: ". . . most ultra-orthodox schools still do not allow military personnel to conduct trainings") should be supported by a proper source. Sometimes it is not clear whether the given statement refers to a direct outcome of this study? For example, lines 395- 398 regarding the "Low Socioeconomic Status". Lines 442 – 444: belong to technology section in lines 481?

**Authors' response:**

Regarding stakeholders:

1. We enclosed the interview guide, which was written to help interviewers frame the interviews and keep them on track, as an appendix. The answers were analyzed thematically and are summarized in clear and straightforward

subsections. Also, they are concluded schematically in the revised SWOT analysis and in points in the conclusion section.

2. The interviewees consisted of 16 relevant national-level policy and decision makers, nine rescue personnel and five religious leaders and key figures in the ultra-orthodox community. Of the interviewees, 17 described themselves as ultra-orthodox but did not highlight which exact community they were affiliated with; the rest were secular or religious but not ultra-orthodox. We added this information to the manuscript.

3. We approached approximately 40 stakeholders, 30 of whom agreed to be interviewed (75% response rate).

All the information that is written in section 4.2 presents qualitative results that we deducted from the interviews with relevant stakeholders. Still, in order to make things clearer, we made the changes recommended by the referee.

- Lines 382-390: We added supporting sources regarding the work of the Home Front Command in the education system. We added quotes from interviewees affiliated with the Home Front Command and with the Ministry of Education supporting our statements:

  **Insularity of Educational Institutions.** In Israel, the military-affiliated Home Front Command invests heavily in providing emergency training to children and youth through schools and other educational institutions (https://www.oref.org.il/11016-he/Pakar.aspx). We learned from our interviews with role holders from Home Front Command and from the Ministry of Education that many institutions in the ultra-orthodox sector do not allow the government or military to enter educational institutions to deliver training even on crucial issues such as emergency preparedness. This is a decisive factor that leaves the ultra-orthodox population less prepared for earthquakes. Gershon from the Home Front Command elaborated on the subject and said: "Everything with relation to the Israeli army, especially in our times, holds a lot of tension. It's complex, very few institutions cooperate with us. The ultra-orthodox education system at large does not allow military elements to enter". Though the Home Front Command has adapted its curriculum to the norms of ultra-orthodox society, most ultra-orthodox schools still do not allow military personnel to conduct training, and therefore the majority of the students do not receive any instruction. Moreover, the Home Front Command has prepared booklets with safety information for students, adapted according to ultra-orthodox norms. According to Ya'ir, a safety officer from the Ministry of Education, many schools will not use these booklets.

- Lines 402-404: The interviewees talked about certain characteristics of the ultra-orthodox community that may create a positive effect on the preparation phase and emergency phase of an earthquake. Lines 402-404 introduce this section.

- Lines 395-398: Quoted in this subsection are the words of the interviewees describing why the low socioeconomic status which characterizes the ultra-orthodox sector may hamper earthquake preparedness.

- In the subsection "Adapting Technologies" we recommended that state authorities change technologies to be religiously appropriate. In the subsection "Soliciting Support" we recommended approaching community leaders and receiving their support in all areas of disaster management, including developing various technologies. The example of sending text messages to call-only phones is not an appropriate technology adaption and therefore requires the support of community leaders. We feel like the right place for this example is in the subsection "Soliciting Support".

**Referee comment: Discussion and Conclusions**

Section 5 is hard to follow. There are issues that belong to the introduction, discussion, conclusions and recommendations, and I got confused which is what. For example, the content in lines 491-495 sounds reasonable, but is it a result of this work or just ideas taken from the given references. Sometimes the phrasing seems to 'hide' the finding of this study, for example, lines 489-491: "Past research has much to say regarding how emergency preparedness among marginalized communities can be improved, including both conduct during emergencies as well as recovery efforts following disasters. Many of our findings support this literature." I think that this should be rephrased the other way around: first state in your own words what you have found and then discuss whether it supports (or not. . .) past research and give the appropriate reference. The main understandings, outcomes and suggestions are scattered along the text and thus I would recommend splitting Section 5 into three: Discussion, Conclusions and Recommendations.

Conclusions

I recommend summarizing the actual findings of this work as independent statements, rather than mentioning that: "Many of our findings support this literature. . .".

Recommendations

Although the recommendations (e.g. lines 549-55) are part of the conclusions, I suggest listing it also in bullets or in a table, so as the authorities, who would (or should) be interested to implement the outcomes of this work, get to the point simply and easily

**Authors' response:**

We thank the referee for this helpful recommendation. As suggested, we split the section "Discussion and Conclusion" into three sections: "Discussion", "Conclusion" and "Recommendations". We thoroughly revised the discussion, stating our findings and then discussing whether they support past research. We reviewed the level of awareness and preparedness of the ultra-orthodox sector and compared the findings with the literature. We discussed the community's unique characteristics and stated how they may increase preparedness or hamper it.

In the discussion, we brought up an interesting point regarding retrofitting: Half of our survey respondents indicated that they do not live in buildings that meet the legal safety standards despite the fact that some of their neighborhoods are in areas with increased ground shaking during an earthquake (Salamon et al., 2010) (Fig. 2). Alarmingly, none of our respondents recommended retrofitting, a crucial strategy for minimizing the harm caused by earthquakes (Bozorgnia and Bertero, 2004). Our interviews with relevant stakeholders further confirmed that very few ultra-orthodox people are interested in retrofitting. They explain that the

reasons for this include a lack of awareness regarding three factors: the potential impact of earthquakes, the significance of building conditions in reducing damage, and the government's willingness to support retrofitting. Additional reasons are the intangibility of the danger and the low economic status of ultra-orthodox society. These findings stand in agreement with Maldonado et al. (2016) and Lucini (2014), who found that minority groups show a low level of self-protection and preparedness, a low level of hazard knowledge and a low level of action during the emergency phase.

In the discussion, we compared the level of knowledge and awareness of the ultra-orthodox population to that of the general Israeli public as stated above.

Lastly, in the discussion we touched upon the fact that in this research we used non-probability sampling that may not represent the entire ultra-orthodox population in Israel.

The research findings are concluded in the "conclusion" section. Our research questions included three main themes: the actual state of earthquake preparedness in the ultra-orthodox sector, characteristics that may hinder or promote preparedness, and ways of improving preparedness. The findings from the first two questions are summarized in a revised SWOT analysis that we further developed. The findings from the third question are summarized in bullets within this section.

As suggested, we listed our recommendations in a separate section using a clear bullet format.

**Referee comment: Discussion and Conclusions**

Oversimplification of the situation? Lines 514-15 state that "Given that the ultra-orthodox constitute a religious minority in Israel, the state authorities' representation of these communities is minimal" The status of the ultra-orthodox minority in Israel is unique. On one hand, they are a significant part of the government, sometimes even the balancing power, and are able to achieve much influence, beyond their relative share in the population. On the other hand, they refrain from accepting the Israeli authorities as is, and adhere to their own community leaders. They seem to prefer to isolate themselves and maintain their own way of life. In that sense, once you conclude and recommend that (line 554-5): "In addition, community leaders should be asked to give their support before approaching the community they lead", it means (in the subtext) that the community leaders may share the responsibility for the ". . . low level of hazard knowledge and a high level of disbelief that a devastating earthquake would occur in their area in the near future." (lines 39-40). As a result, a possible recommendation could be that instead of authorizing ". . . local non-governmental organizations official responsibilities in the area of earthquake. . . " (line 553), the local community leaders should authorize the formal Israeli authorities the responsibilities in the area of earthquake. This is a very complex situation and it means that improving the level of knowledge and awareness in this minority group is not just a simple list of 'to do'.

**Authors' response:**

We appreciate the referee's criticism regarding the oversimplification of the situation. We deleted the cited sentence in lines 514-515 regarding the minimal representation of the ultra-orthodox communities in state authorities. In the revised manuscript, the presentation of the complex relationship between the state and the ultra-orthodox society is more subtle and nuanced. Indeed, as the referee points out, this relationship often includes a high level of suspicion by ultra-orthodox communities towards state institutions and authorities, as well as

these communities' preference to adhere to their own community leaders. Moreover, we agree with the referee that the local community leaders share the responsibility for the current low level of hazard knowledge. We now emphasize the urgent need to establish a reciprocal and continuing dialogue and collaboration between the relevant state institutions and ultra-orthodox community leaders. For this purpose, both parties should acknowledge the necessity to build trust and joint work networks to improve the level of preparedness. We demonstrate the required change by showing how dozens of our ultra-orthodox students currently serve as trainers and "ambassadors" in their own communities, providing culturally-adapted knowledge on earthquake preparedness and receiving the support of key actors in their ultra-orthodox communities. We hope that the revised manuscript better reflects the complex and entangled situation highlighted by the referee.

**Referee comment: Referencing**

Many statements sounds reasonable, however, they need to be supported by a reference or source of data, e.g. line 90; lines 209-210: "With regard to earthquake preparedness, most members of the Jewish ultra-orthodox society in Israel live in . . . buildings that do not meet the standards for earthquakes."

**Authors' response:**

We added supporting references to various statements and deleted some other statements. For example, for the statement in lines 209-210, we now provide the Central Bureau of Statistics' data regarding the density of the Jewish ultra-orthodox population (1.37 persons per room) in comparison to the Jewish secular population (0.72 persons per room), and we demonstrate that many ultra-orthodox neighborhoods, e.g. in Jerusalem and Bnei Brak, had been built before the current seismic building code was implemented.

Line 90, "marginalized social and cultural groups are more vulnerable to natural disasters than majority groups", has been removed from the revised literature review which now focuses more on religion and less on marginalization, following Referee 1's suggestion.

**Referee comment: References not mentioned**

Avni, R., 1999. The 1927 Jericho Earthquake, Comprehensive Macroseismic Analysis Based on Contemporary Sources. (PhD.). Ben Gurion University, Beer-Sheva. Salamon, A., Katz, O. and Crouvi, O. (2010). Zones of required investigation for earthquake-related hazards in Jerusalem. Natural Hazards, 53: 375-406. Zohar, M., Salamon, A. and Rubin, R., (2017). Earthquake damage history in Israel and its close surrounding - evaluation of spatial and temporal patterns. Tectonophysics, 696: 1-13.

**Authors' response:**

We thank the referee for suggesting these references. The detailed information of Avni, 1999 (written in Hebrew) had been translated and summarized at our initiative in Hough & Avni (2011) which we cited. In the revision we use it together with the work of Salamon et al. (2010) and cite both in the references to the text and figures.

To conclude, we addressed all the issues raised by Referee 2. We would like to thank the referee again for the work invested in reviewing our manuscript. We truly believe that the review process substantially contributed to the article and we hope that the current version will be accepted to *Natural Hazards and Earth System Sciences*.

Sincerely,

The authors

**Figures**

[Figure]

**Figure 1:**

**(a)- Estimated spatial extent of ruptures from historic periods along the DSFS (after Agnon, 2014). The position of the events in the graph is projected from the map of the DSFS below in (b).**

**(b) - The DSFS main branches (Hamiel et al., 2018; Hofstetter et al., 1996; Kagan et al., 2011; Politi, 2011; Sharon et al., 2018) over instrumental earthquakes record (from 1984) of >3 Md (www.gii.co.il).**

**Settlements with Ultra-Orthodox residents are marked; the majority is in Jerusalem and Bnei Brak. [1] – Beit Shemesh, [2] – Beitar Illit, [3] – Modi'in Illit.**

[Figure]

**Figure 2:**

**Map of neighborhoods in Jerusalem with Ultra-Orthodox residents (modified after Community administrators Jerusalem map, 2009; Golan, 2020) over ground amplification map (Salamon et al., 2010).**

[1] – Har Nof, [2] – Bait Vegan, [3] _ Kiryat Hayovel, [4] – Ramot, [5] – Romema, [6] - Givat Mordechai, [7] - French Hill, [8] - Mekor Baruch, [9] - Ramat Eshkol, [10] - Sha'arei Hesed, [11] - Zikhron Moshe, [12] – Katamon and Katamonim, [13] – Arnona, [14] – Bukharim, [15] – Morasha, [16] - Shmuel HaNavi.

**Mea Shearim, and the neighborhoods adjacent to it to the north are associated with the ultra-orthodox sector. They are in an area marked by increased ground shaking during an earthquake.**

**Instrumental earthquakes (from 1984) of <2 Md are presented by black rectangles together with the year of quaking ([www.gii.co.il](http://www.gii.co.il)). Medvedev–Sponheuer–Karnik scale (MSK) intensity scale evaluates the severity of ground shaking in an area of earthquake occurrence; the mean seismic intensity in Jerusalem shows strong-to very strong shaking zones from the 1927 6.2 M Jericho earthquake (Avni, 1999).**

[Figure]

Level of belief that a disastrous earthquake will occur in Israel/in your area in the next five years

**Figure 3:**
Level of belief that a disastrous earthquake will occur in Israel/in your area in the next five years.

[Figure]

**Figure 4:**
Level of knowledge and readiness to prepare for an earthquake in Israel

---

## Author Response (AR1)

September 23, 2020

**Authors' response to editor's and referees' comments**

Enclosed, please find a revised version of our manuscript, "Earthquake preparedness among religious minority groups: The case of the Jewish ultra-Orthodox society in Israel." We would like to take this opportunity to thank the Handling Editor and the referees for the exceptionally constructive feedback. We found the comments and suggestions that they offered very helpful and we appreciate the opportunity to revise key aspects of the article. We have made all of the changes suggested by the Handling Editor and referees, and we believe that the revisions create a stronger and clearer paper. We highlighted the main changes to the manuscript within the document by using colored (blue) text. To facilitate review of the revised manuscript, we outline the main changes that were made:

**Authors' response to the Handling Editor**

**Editor comment: Figure 1b**

Figure 1b (the revised version) shows the 1927 Dead Sea EQ (?) which is not postdating 1984 as indicated in the caption. In addition, what is the population-threshold/criteria for a settlement to be presented in the figure as 'with UO residence'? Please state clearly.

**Authors' response:**

We fixed the caption to indicate that earthquakes are presented from 1900.

The criterion for a settlement to be presented in the figure is a settlement with over a thousand ultra-Orthodox residents, according to Shahak, 2017. We clarified this criterion in the figure's caption.

**Editor comment: Figure 2**

Figure 2 (the revised version). Similar to the above, what is the population-threshold/criteria for a neighborhood to be presented in the figure as 'with UO residence'? Please state clearly. E.g. Kiryat Yovel/Arnona are different from Har Nof. The figure caption as a whole is not clear, please revise.

Why showing the instrumental <2 Md earthquakes? They are irrelevant to the seismic risk, are not the one presented in Fig. 1a and are not defining seismic zones. Consider omitting them. I am not sure the map in the background is the one to present. It shows local geological conditions of ground amplification.

**Authors' response:**

We revised the figure to include only ultra-Orthodox neighborhoods in Jerusalem. An ultra-Orthodox neighborhood is defined as a neighborhood where over 50% of the

Jewish population aged 15 or older are ultra-Orthodox (Korach and Choshen, 2020, p. 25). This criterion is now included in the figure's caption.

The figures' captions were edited.

Following the suggestion, the M<2 earthquakes were removed and only a single M>3 earthquake is presented. This earthquake is sufficiently large to be felt.

The aim of showing a ground amplification map is to emphasize that in some ultra-Orthodox neighborhoods there may be excessive shaking that could risk the residents.

**Authors' response to Referee 1**

**Referee comment: Abstract**

The abstract outlines the findings of the survey and interviews, but the last sentence does not outline any new idea.

**Authors' response:**

We concluded the abstract by stating the uniqueness of this research: it studies a religious group that is also a cultural minority and therefore requires special adaptations. This research offers a perspective on the complex reality of hazard preparedness in a religiously diverse country and its conclusions are applicable to other countries and natural hazards. We also listed some of the key adaptations suggested.

**Referee comment: Introduction**

Please rewrite with more specific and more precise data and observations.

**Authors' response:**

We rewrote the introduction using the following structure: information on earthquake destruction; the significance of the research topic and the research gap; the goal of our research; the research questions; summary of the main argument. We believe that this new structure reflects more precisely the main issues that are discussed in the article.

**Referee comment: Earthquake preparedness among marginalized minority groups and religious communities**

Due to the title of the paper, this part should be rewritten to focus only on religious communities of the world or of the region, and not on 'marginalized minority groups'.

**Authors' response:**

We rewrote this section (2.1) focusing on disaster preparedness in religious communities, as described in past literature. We discussed the impact of religion on the three phases of an earthquake: the preparation, emergency, and restoration phases. We included suggestions from past literature on how to improve the response of religious communities during these stages. We used additional sources such as Ngin et al., 2020; Waitzberg et al., 2020; Wilkin et. al, 2019, among others.

Since the social group that we study is also a minority in its country, we briefly listed several characteristics of minorities that intersect with religion and religiosity and have a great impact on disaster preparedness.

**Referee comment: The Jewish Ultra-Orthodox Sector**

Does the word "sector" mean that the JUO community live in a specific area? As it is not well known in Europe for instance, please also precise if the JUO have a specific language, etc.

**Authors' response:**

Most ultra-Orthodox people live either in towns of their own or in closed community neighborhoods within diverse cities.

Most of the ultra-Orthodox people are Hebrew speakers, although some of the communities still speak Yiddish as their first language.

We added this information in section 2.2.

**Referee comment: Earthquake Hazard in Jerusalem**

Please add a figure showing Israel, Jerusalem, and fault lines with dots or stars illustrating historic and instrumental earthquakes with intensities or magnitudes. Make a zoom on the eventual JUO sector of Jerusalem and eventual seismic zoning (if it does exist) and eventual shaking maps (if they do exist): the aim of this demand is to illustrate the specific vulnerability of the people and buildings

**Referee comment: Figure 1**

in an inset, show map of Israel + Jerusalem location + any specific JUO sector Table 1: precise 'epicentral' or 'maximal' or 'downtown' or 'whatever' intensity

**Authors' response:**

We added two figures that demonstrate the specific vulnerability of the people and buildings of the ultra-Orthodox society:

Figure 1 - instrumental earthquakes (M>3) were added (Figure 1b); historical earthquakes are shown in Figure 1a. In Figure 1b the dominant faults are marked, as well as settlements with significant ultra-Orthodox population.

Figure 2 - a map of Jerusalem with ultra-Orthodox neighborhoods, calculated seismic intensities (Avni, 1999), zones of predicted ground amplification (Salamon et al., 2010), and instrumental earthquakes M>3 in Jerusalem.

**Referee comment: Survey**

How many questionnaires were sent OR give a percentage of responses (can help distinguishing persons interested in preparedness who could later be used as vectors of information), see line 337

**Authors' response:**

The response rate was around 90% for the in-person interviews. The response rate for the online questionnaire is undefined since it was distributed via a free link. We added this information in section 3.1.

**Referee comment: Results**

Please simplify and re-order the answers.

**Authors' response:**

We revised the results section – including quantitative and qualitative results – to make this section clearer. We simplified and re-ordered answers and elaborated on the demographics of the respondents. We added data regarding the community's level of preparedness, the negative and positive effects of religion on preparedness, and ways to improve preparedness. Furthermore, we added the translated quantitative questionnaire (Appendix A), qualitative interview guide (Appendix B), and tables with quantitative findings (Appendix C). Consequently, we believe that it is now much easier to follow the results.

**Referee comment: Quantitative findings**

Help the reader with graphs, pie charts etc.

**Authors' response:**

We included diagrams to illustrate the main findings (Charts 1-3). For instance, we added a diagram to summarize the respondents' level of belief regarding the occurrence of a disastrous earthquake in the near future (Chart 1).

**Referee comment: Qualitative findings**

- Outline the answers of the religious leaders and key figures of the JUO community, for they could become vectors of preparedness information.

- Family names of safety officers are cited, is it common and accepted by the so-called persons, and is it allowed by the NHESS journal?

**Authors' response:**

- In the findings section, we added more empirical data based on the answers of religious leaders, rabbis, and educators, particularly in the context of how they could become vectors of preparedness information (sections 4.2.1, 4.2.2, 4.2.3). Further, the answers of the respondents are summarized in a revised SWOT analysis (Table 3) and in clearer and revised points in the conclusion section. Our recommendations are also deducted from the answers of the religious leaders and key figures that we interviewed.

- All the research participants' names in the manuscript are pseudonyms. We clarified this in section 3.2.

**Referee comment: Discussion and Conclusions**

The discussion needs to be reworked, also focusing on religious minorities. The SWOT analyze is a very good idea that could be more developed to offer preparedness perspectives based on the strong social capital of the JUO. A good perspective would be to submit preparedness ideas to interviewed religious leaders: then emphasis could be put on advices in hardware stores, simple and cheap home security (furniture organization and securing, basic rescue kits for the family), practical skills based on the specific neighborhood solidarity in the JUO, the organization of participatory experiences by the religious authorities for instance.

**Authors' response:**

We revised the discussion focusing on religious minorities as suggested. We reviewed the level of awareness and preparedness of the ultra-Orthodox society and compared the findings to the literature on other religious groups. We discussed the community's unique characteristics and explained how they may increase preparedness or hamper it. We also compared the ultra-Orthodox population's level of knowledge and readiness to prepare for an earthquake to that of the general Israeli public (Ya'ar et al., 2015) (Chart 3).

We developed the SWOT analysis in accordance with the referee's recommendation, emphasizing the potential contribution of the ultra-Orthodox society's strong social capital. We elaborated on this potential contribution in the findings section as well (in section 4.2.3: "Leveraging Solidarity"). We thank the reviewer for the useful ideas for upgrading preparedness that we incorporated in the manuscript.

**Authors' response to Referee 2**

**Referee comment: Earthquake Hazard in Jerusalem**

Section 2.3 does not present the state-of-the-art knowledge and understandings regarding earthquake hazards and damage in Jerusalem. For example, Avni (1999) presented detailed lists and reports of what had happened in Jerusalem during the 1927 earthquake; Salamon et al. (2010) conducted earthquake hazard evaluation for Jerusalem; Zohar et al. (2017) presented updated list of earthquakes and the damage they cause in Israel, including in Jerusalem. With this respect, Section 2.3 and Table 1 should be revised thoroughly.

**Authors' response:**

The detailed information of Avni, 1999 (written in Hebrew) had been translated and summarized at our initiative in Hough & Avni (2011) which we cited. Following the referees' suggestions, we use some of the detailed information from Avni (1999) together with the work of Salamon et al. (2010). The manuscript was updated using the recommended articles:

Figure 1 - instrumental earthquakes (M>3) were added (Figure 1b); historical earthquakes are shown in Figure 1a. In Figure 1b the dominant faults are marked, as well as settlements with significant ultra-Orthodox population.

Figure 2 - a map of Jerusalem with ultra-Orthodox neighborhoods, calculated seismic intensities (Avni, 1999), zones of predicted ground amplification (Salamon et al., 2010), and instrumental earthquakes M>3 in Jerusalem.

**Referee comment: Survey**

- Lines 256-7: "We then created a shortened version of the questionnaire to spread via an online form." However, in lines 371-2 it is said: "Most people do not own a television at home, and many do not have access to the internet or the radio." The reader may understand that those who replied online do not represent that group?

- Likewise, there is a need to show that the group of interviewees (228 responses) does represent the ultra-Orthodox group (one million), so that the results of this study can be generalized and represent the group.

**Authors' response:**

- Regarding internet access, more than half of the respondents had no smartphone or had a smartphone with limited accessibility to internet content for religious reasons (Table 2). Of those who did not own a smartphone, 69% answered the questionnaire via in-person interview. However, 31% of the non-smartphone owners answered the online form; this implies that many ultra-Orthodox people have (albeit limited) internet access. It should be noted that limited internet access was one of the considerations for choosing in-person interviews for most of the respondents. According to Cahaner and Malach (2019, p. 69), 49% of the ultra-Orthodox adults use the internet, in comparison to 89% among non-ultra-Orthodox Jews in Israel. This information was added in section 4.1.1.

- In the survey, we used non-probability sampling that may not represent the entire ultra-Orthodox population in Israel. There were only 288 respondents, and there was an over-representation of men. The average age was young, and a great majority of respondents lived in Jerusalem and its surrounding areas. However, in our opinion, our sample also has considerable advantages. First, the in-person interview survey was conducted by ultra-Orthodox or Orthodox surveyors, enabling us to reach out to ultra-Orthodox people who would probably refuse to respond to a telephone survey from an academic institution. A telephone survey, which is faster and simpler, has a lower response rate and might involve considerable bias when it comes to the ultra-Orthodox public. Second, the survey conducted in face-to-face interviews allowed a very long and detailed questionnaire to be answered. It allowed the elaboration of questions and answers, ensured that the questions and answers were understood, and avoided offhand answers. None of this happens in a telephone or internet survey. Third, focusing on particular communities, such as the ultra-Orthodox communities in Jerusalem and its surrounding areas, allows a deeper understanding of their needs and challenges and enables the possibility of working with them over time. For example, it allows efficient training in these communities, like the training led by dozens of our ultra-Orthodox students in several ultra-Orthodox communities in Jerusalem, after we trained them (as now indicated in sections 3.2 and 4.2.3: Adapting the State's Trainings). Last, the young people interviewed either had or were about to have children. We have learned from the literature that children's education is one of the most impactful tools in disaster risk reduction. Hence, we wanted to work with these young people. This explanation now appears in the Discussion (section 5).

**Referee comment: Results (general comment)**

Overall, the raw data is not presented and the little that is shown does not allow the reader to follow and repeat the results and the final conclusions. The raw data should be presented in full, in tables and diagrams.

**Author response:**

Please see the two subsequent responses below.

**Referee comment: Quantitative findings**

Quantitative analysis:

1. There is a need to present the online questionnaire in full, maybe as an appendix.

2. Please present the demography of those who responded and filled the questionnaire, i.e. the information relevant to this study such as age, gender, income, education, area of living, social subgroup, etc. This of course should be kept within the confidentiality and anonymity promised to the reviewees, but still allow sufficient transparency of the database. Section 4.1.1 is not enough.

3. Please report how many forms were distributed against how many were replied and filled.

4. The findings from the questionnaires should also be presented in full as a background for the analysis of the results and the discussion.

5. It is not possible to evaluate how much the level of knowledge and awareness of the ultra-Orthodox group differs from the 'reference standard' of Israel. There is a need to present the 'common' level of knowledge in Israel, and by comparison, show the difference (sections 4.1.2, 4.1.3).

**Authors' response:**

We thank the referee for this helpful comment; we have added all that was recommended.

We included diagrams to illustrate the main findings (Charts 1-3). For instance, we added a diagram to summarize the respondents' level of belief regarding the occurrence of a disastrous earthquake in the near future (Chart 1).

We updated and elaborated on the demographics of the respondents (age, gender, marital status, children, economic status, education, area of living, social subgroup, and type of mobile phone they own). This information appears in section 4.1.1 and Table 2.

We added substantial quantitative and qualitative data regarding the community's level of preparedness. Moreover, following the referee's advice, we added the questionnaire in Appendix A and the quantitative findings in Appendix C.

The response rate was around 90% for the in-person interviews. The response rate for the online questionnaire is undefined since it was distributed via a free link. We added this information in section 3.1.

In the Discussion and in Chart 3, we compared the ultra-Orthodox population's level of knowledge and readiness to prepare for an earthquake to that of the general Israeli public as found by Ya'ar et al. (2015).

**Referee comment: Qualitative findings**

1. Even though the interviews were open, there is a need to show the main leading issues and questions, as well as the answers, at least in a schematic way. Otherwise, there is no way to infer and conclude the systematic attitude or approach of the examined population.

2. The demography of those who participated in the open interview should also be presented. Which segment of the ultra-Orthodox group do they belong to? Does this segment represent the entire ultra-Orthodox group?

3. Please note how many people were approached against how many were interviewed.

Some sections or paragraphs seem to belong to the Introduction rather than to the results, e.g. lines 382-390, 402-404. In any case, such facts or statements (e.g. lines 389– 90: ". . . most ultra-Orthodox schools still do not allow military personnel to conduct trainings") should be supported by a proper source. Sometimes it is not clear whether the given statement refers to a direct outcome of this study? For example, lines 395- 398 regarding the "Low Socioeconomic Status." Lines 442 – 444: belong to technology section in lines 481?

**Authors' response:**

Regarding stakeholders:

1. We enclosed the interview guide (qualitative questionnaire) as Appendix B. The answers were analyzed thematically and are now summarized in clear and straightforward subsections in section 4.2. We incorporated additional representative qualitative data (citations from interviews and focus groups) to allow an in-depth understanding of the research participants' approach. Also, their answers are concluded schematically in the revised SWOT analysis and in points in the Conclusion section.

2. The interviewees consisted of 16 relevant national-level policy and decision makers (e.g., in the Home Front Command, Ministry of Health, Ministry of Education, National Steering Committee for Earthquake Preparedness, etc.), 10 officials in rescue organizations, and five religious leaders and key figures in the ultra-Orthodox community.

Of the interviewees, 17 described themselves as ultra-Orthodox but did not specify the community they were affiliated with; the rest were secular or religious but not ultra-Orthodox.

In addition, focus groups were held with the rescue organization Magen David Adom, ultra-Orthodox educators and teachers, and an ultra-Orthodox organization for people with disabilities.

We included this information in section 3.2.

3. We approached approximately 41 stakeholders, 31 of whom agreed to be interviewed (75% response rate).

All the information that is written in section 4.2 is based on the qualitative results deducted from the interviews and focus groups with relevant stakeholders. Still, in order to clarify the results, we made the changes recommended by the referee:

- Section 4.2.1, paragraph on Insularity of Educational Institutions: We added supporting sources regarding the work of the Home Front Command in the education system. We added quotes from interviewees affiliated with the Home Front Command and the Ministry of Education supporting our statements.

- Lines 402-404 (in the previous version): The interviewees talked about characteristics of the ultra-Orthodox community that may create a positive effect on the preparation and emergency phases of an earthquake. Lines 402-404 introduce this section.

- Lines 395-398 (in the previous version): This paragraph reflects the assertion of the interviewees describing why the low socioeconomic status that characterizes the ultra-Orthodox society may hamper earthquake preparedness. We edited this paragraph to clarify this point.

- In the subsection "Adapting Technologies" we recommended that state authorities change technologies to be religiously appropriate. In the subsection "Soliciting Support" we recommended approaching community leaders and receiving their support in all areas of disaster management, including developing various technologies. The example of sending text messages to call-only phones is not a religiously appropriate technology and therefore requires the support of community leaders. We feel like the right place for this example is in the subsection "Soliciting Support."

**Referee comment: Discussion and Conclusions**

Section 5 is hard to follow. There are issues that belong to the introduction, discussion, conclusions and recommendations, and I got confused which is what. For example, the content in lines 491-495 sounds reasonable, but is it a result of this work or just ideas taken from the given references. Sometimes the phrasing seems to 'hide' the finding of this study, for example, lines 489-491: "Past research has much to say regarding how emergency preparedness among marginalized communities can be improved, including both conduct during emergencies as well as recovery efforts following disasters. Many of our findings support this literature." I think that this should be rephrased the other way around: first state in your own words what you have found and then discuss whether it supports (or not. . .) past research and give the appropriate reference. The main understandings, outcomes and suggestions are scattered along the text and thus I would recommend splitting Section 5 into three: Discussion, Conclusions and Recommendations.

Conclusions

I recommend summarizing the actual findings of this work as independent statements, rather than mentioning that: "Many of our findings support this literature. . .".

Recommendations

Although the recommendations (e.g. lines 549-55) are part of the conclusions, I suggest listing it also in bullets or in a table, so as the authorities, who would (or should) be interested to implement the outcomes of this work, get to the point simply and easily

**Authors' response:**

We thank the referee for this helpful recommendation. As suggested, we split the section "Discussion and Conclusion" into three sections: "Discussion," "Conclusion," and "Recommendations." We thoroughly revised the discussion, stating our findings and then discussing whether they support past research. We reviewed the level of awareness and preparedness of the ultra-Orthodox society and compared the findings to the literature, including to a previous study regarding the general Israeli population. We discussed the community's unique characteristics and stated how they may increase preparedness or hamper it.

The research findings are concluded in the "Conclusion" section. Our research questions included three main themes: the actual state of earthquake preparedness in the ultra-Orthodox society, characteristics that may hinder or promote preparedness, and ways of improving preparedness. The findings from the first two questions are summarized in a revised SWOT analysis that we further developed (Table 3). The findings from the third question are summarized in bullets in the "Conclusion" section.

As suggested, we listed our recommendations in a separate section (7) using a clear bullet format.

**Referee comment: Discussion and Conclusions**

Oversimplification of the situation? Lines 514-15 state that "Given that the ultra-Orthodox constitute a religious minority in Israel, the state authorities' representation of these communities is minimal." The status of the ultra-Orthodox minority in Israel is unique. On one hand, they are a significant part of the government, sometimes even the balancing power, and are able to achieve much influence, beyond their relative share in the population. On the other hand, they refrain from accepting the Israeli authorities as is, and adhere to their own community leaders. They seem to prefer to isolate themselves and maintain their own way of life. In that sense, once you conclude and recommend that (line 554-5): "In addition, community leaders should be asked to give their support before approaching the community they lead," it means (in the subtext) that the community leaders may share the responsibility for the ". . . low level of hazard knowledge and a high level of disbelief that a devastating earthquake would occur in their area in the near future." (lines 39-40). As a result, a possible recommendation could be that instead of authorizing ". . . local non-governmental organizations official responsibilities in the area of earthquake. . ." (line 553), the local community leaders should authorize the formal Israeli authorities the responsibilities in the area of earthquake. This is a very complex situation and it means that improving the level of knowledge and awareness in this minority group is not just a simple list of 'to do.'

**Authors' response:**

We appreciate the referee's criticism regarding the oversimplification of the situation. We deleted the cited sentence in lines 514-515 (in the previous version) regarding the minimal representation of the ultra-Orthodox communities in state authorities. In the revised manuscript, the presentation of the complex relationship between the state and the ultra-Orthodox society is more subtle and nuanced. Indeed, as the referee points out, this relationship often includes a high level of suspicion by ultra-Orthodox communities towards state institutions and authorities, as well as these communities' preference to adhere to their own community leaders. Moreover, we agree with the referee that the local community leaders share the responsibility for the current low level of hazard knowledge, and this is now incorporated in the Discussion.

In the revised manuscript we emphasize the urgent need to establish a reciprocal and continuing dialogue and collaboration between the relevant state institutions and ultra-Orthodox community leaders. For this purpose, both parties should acknowledge the necessity to build trust and joint work networks to improve the level of preparedness. The lack of trust and collaboration have had severe ramifications during the COVID-19 pandemic. Furthermore, we now quote our research participants' opinion regarding the impact that a joint statement of rabbis and experts on earthquakes may have on the preparedness of the ultra-Orthodox society. Based on the qualitative results, we propose that a delicate balance seems to be needed between the ultra-Orthodox community autonomy, on the one hand, and supervision and a strict professional standard defined by state authorities, on the other hand.

We hope that we now better reflect the complexity and entanglement of the situation in the sections Results, Discussion, Conclusion, and Recommendations.

**Referee comment: Referencing**

Many statements sounds reasonable, however, they need to be supported by a reference or source of data, e.g. line 90; lines 209-210: "With regard to earthquake preparedness, most members of the Jewish ultra-Orthodox society in Israel live in . . . buildings that do not meet the standards for earthquakes."

**Authors' response:**

We added supporting references to various statements and deleted some other statements. For example, for the statement in lines 209-210 (in the previous version), we now provide the Central Bureau of Statistics' data (2019) regarding the density of the Jewish ultra-Orthodox population (1.35 persons per room) in comparison to the density of the Jewish secular population (0.71 persons per room).

Line 90 (in the previous version), "marginalized social and cultural groups are more vulnerable to natural disasters than majority groups," has been removed from the revised literature review which now focuses more on religion and less on marginalization, following Referee 1's suggestion.

**Referee comment: References not mentioned**

Avni, R., 1999. The 1927 Jericho Earthquake, Comprehensive Macroseismic Analysis Based on Contemporary Sources. (PhD.). Ben Gurion University, Beer-Sheva. Salamon, A., Katz, O. and Crouvi, O. (2010). Zones of required investigation for earthquake-related hazards in Jerusalem. Natural Hazards, 53: 375-406. Zohar, M., Salamon, A. and Rubin, R., (2017). Earthquake damage history in Israel and its close surrounding - evaluation of spatial and temporal patterns. Tectonophysics, 696: 1-13.

**Authors' response:**

We thank the referee for suggesting these references. The detailed information of Avni, 1999 (written in Hebrew) had been translated and summarized at our initiative in Hough & Avni (2011) which we cited. Following the referees' suggestions, we use some of the detailed information from Avni (1999) together with the work of Salamon et al. (2010) and cite both in the references to the text and Figure 2.

**Conclusion**

To conclude, we addressed all the issues raised by the Handling Editor and the two referees. We would like to thank them again for the work they invested in reviewing our manuscript. We truly believe that the review process substantially contributed to the article and we hope that the current version will be accepted to *Natural Hazards and Earth System Sciences*.

Sincerely,

The authors

[revised manuscript text omitted]